# MAD2L2 promotes replication fork protection and recovery in a shieldin-independent and REV3L-dependent manner

Inés Paniagua [1], Zainab Tayeh[1], Mattia Falcone [1], Santiago Hernández Pérez [1], Aurora Cerutti[1] & Jacqueline J. L. Jacobs [1] ✉

Protection of stalled replication forks is essential to prevent genome instability, a major driving force of tumorigenesis. Several key regulators of DNA double-stranded break (DSB) repair, including 53BP1 and RIF1, have been implicated in fork protection. MAD2L2, also known as REV7, plays an important role downstream of 53BP1/RIF1 by counteracting resection at DSBs in the recently discovered shieldin complex. The ability to bind and counteract resection at exposed DNA ends at DSBs makes MAD2L2/shieldin a prime candidate for also suppressing nucleolytic processing at stalled replication forks. However, the function of MAD2L2/shieldin outside of DNA repair is unknown. Here we address this by using genetic and single-molecule analyses and find that MAD2L2 is required for protecting and restarting stalled replication forks. MAD2L2 loss leads to uncontrolled MRE11-dependent resection of stalled forks and single-stranded DNA accumulation, which causes irreparable genomic damage. Unexpectedly, MAD2L2 limits resection at stalled forks independently of shieldin, since fork protection remained unaffected by shieldin loss. Instead, MAD2L2 cooperates with the DNA polymerases REV3L and REV1 to promote fork stability. Thus, MAD2L2 suppresses aberrant nucleolytic processing both at DSBs and stalled replication forks by differentially engaging shieldin and REV1/REV3L, respectively.

The transmission of genetic material to daughter cells depends on the faithful completion of DNA replication during S-phase. However, DNA replication is frequently challenged by endogenous and exogenous stresses that can lead to replication fork slowdown and/or stalling[1]. Persistent replication fork stalling can ultimately result in fork collapse and the formation of DNA double-stranded breaks (DSBs), which are lethal if left unrepaired[2]. Thus, strict regulation of DNA synthesis and the DNA damage response is crucial to maintain genome integrity. Multiple pathways ensure replication fork progression under stress, including translesion DNA synthesis (TLS), replication repriming, and fork reversal[1]. The latter one involves the conversion of a typical replication fork into a four-way structure to promote fork stability[3,4], albeit exposing DNA ends that are susceptible to nucleolytic attack[5–7].

Protection of reversed forks is thus necessary to prevent excessive degradation of nascent DNA. Consequently, in the absence of key protective factors such as BRCA1/2, FANCD2, FANCA and ABRO1, nascent DNA degradation results in genomic instability, an enabling hallmark of cancer[8–11]. Notably, chemoresistance has been linked to the restored ability of e.g. BRCA1-deficient cells to protect forks from degradation through mechanisms that remain unclear[7,10,12]. Therefore, understanding how cells protect stalled replication forks and promote their recovery is critical for developing and improving strategies for cancer treatment.

MAD2L2 is a small HORMA domain protein with no known catalytic activity that plays an important role in DNA damage tolerance as well as in DSB repair by functioning in two distinct complexes: DNA

[1]Division of Oncogenomics, The Netherlands Cancer Institute, Plesmanlaan 121, 1066 CX Amsterdam, The Netherlands. ✉e-mail: j.jacobs@nki.nl

polymerase ζ (Polζ, composed of MAD2L2 and REV3L) and shieldin (composed of MAD2L2, SHLD1, SHLD2, and SHLD3)[13,14]. On the one hand, MAD2L2 facilitates replication bypass of damaged DNA through TLS, by bridging the interaction between the REV3L polymerase subunit of Polζ and its upstream recruiter REV1[15–17]. On the other hand, MAD2L2 promotes the non-homologous end-joining (NHEJ) DNA repair pathway in the recently discovered shieldin complex, by counteracting resection at broken DNA ends downstream of the DNA damage response factors 53BP1 and RIF1[18–27]. Interestingly, several suppressors of DSB resection also limit nuclease activities at stalled replication forks. For instance, 53BP1 has been described to protect reversed forks from DNA2-mediated degradation[28], although this role is cell-type specific[10,28–32]. Meanwhile, RIF1, the downstream effector of 53BP1 in NHEJ repair, acts independently in the protection of stalled replication forks by controlling the DNA2 nuclease and promoting efficient fork restart[32–34]. Moreover, the CST complex (CTC1-STN1-TEN1), a downstream factor of 53BP1/RIF1 and shieldin in end resection suppression, localizes at stalled forks to prevent uncontrolled degradation by MRE11[35]. However, the involvement of MAD2L2/shieldin in this context has not been characterized. Here we reveal that MAD2L2 is essential for the protection of stalled replication forks and their efficient restart after replication stress. Unexpectedly, this function of MAD2L2 at replication forks is independent of its interaction with shieldin, but instead requires the TLS factors REV3L and REV1. Collectively, our findings identify an unanticipated role for MAD2L2/REV3L/REV1 in genome stability maintenance via the protection of replication forks under conditions of replicative stress.

## Results

### MAD2L2 promotes DNA replication during physiological and exogenous replication stress

To elucidate the role of MAD2L2 in DNA replication, we assessed the proliferation and survival of MAD2L2-depleted cells exposed to different replication poisons. Interestingly, shRNA-mediated MAD2L2 depletion sensitized RPE1-hTERT TP53-/- cells to agents that inhibit DNA polymerase α (aphidicolin) or reduce the cellular nucleotide pool available for DNA synthesis (hydroxyurea (HU))[36], but do not elicit DNA lesions (Supplementary Fig. 1a–c). This is surprising because MAD2L2 activity in replication is thought to be restricted to replication bypass of damaged DNA templates via TLS. However, since MAD2L2 depletion already impaired cell proliferation in the absence of replication poisons, this complicated appreciation of how much replication issues contribute to the impaired proliferation of MAD2L2-depleted cells treated with HU or aphidicolin. To address this differently, we generated inducible MAD2L2 knockout HeLa cells using a doxycycline-inducible CRISPR-Cas9 system. In agreement with previous studies showing that MAD2L2 is essential for interstrand crosslink repair[37,38], MAD2L2 depletion sensitized HeLa cells to the DNA cross-linking agents cisplatin (CIS) and mitomycin C (MMC) (Fig. 1a, b, Supplementary Fig. 1d). Additionally, we measured global DNA synthesis in MAD2L2-depleted HeLa cells by pulse-labeling S-phase cells with the thymidine analog 5-ethynyl-2′-deoxyuridine (EdU). In line with their observed sensitivity to MMC, EdU incorporation was dramatically reduced in MAD2L2-depleted Hela cells treated with MMC (Fig. 1c, d). However, also unchallenged MAD2L2-depleted cells showed reduced EdU incorporation compared to control cells, indicating that MAD2L2 impacts global replication also in unperturbed conditions (Fig. 1c, d). Taken together, these results indicate that MAD2L2 plays a role in protecting cells from replicative stress that is broader than is expected from a TLS factor promoting replication past replication blocking DNA lesions.

### MAD2L2 localizes to stalled replication forks and promotes their progression

To address if MAD2L2 associates with replication forks, we performed the quantitative in situ analysis of protein interactions at DNA

replication forks (SIRF) assay. This technique uses sensitive proximity ligation chemistry to detect protein interactions with nascent DNA at single-cell resolution[39]. Briefly, HeLa cells were incubated with EdU to label nascent DNA, treated with 4 mM HU for 2 h to induce fork stalling, and subsequently processed for SIRF analysis (Fig. 1e). We included an MRE11-EdU SIRF experiment as a positive control. As expected, MRE11 was significantly enriched at nascent DNA following HU treatment (Fig. 1f), consistent with its role at stalled replication forks[7]. Importantly, we detected the association of endogenous MAD2L2 to nascent DNA in untreated conditions, which was significantly increased upon HU treatment (Fig. 1g). These data indicate that MAD2L2 localizes to both active and stalled replication forks.

To address the mechanism by which MAD2L2 promotes DNA replication, we analyzed individual replication forks using DNA fiber assays[40,41] (Fig. 2a). First, we quantified fork progression rates in MAD2L2-depleted HeLa cells by pulse-labeling cells with chlorodeoxyuridine (CldU) for 30 min, followed by a 1 h labeling period with iododeoxyuridine (IdU) in the presence or absence of 4 mM HU. The length of the IdU tracts relative to the labeling time was measured as a readout of replication fork progression. We observed a statistically significant reduction in fork progression in unchallenged MAD2L2-depleted cells (Fig. 2b, c), mimicking the global arrest of DNA replication that we observed by assessing EdU incorporation (Fig. 1d). These results suggest that MAD2L2 loss alone leads to increased fork stalling or fork collapse. Consistently, short exposure to HU further impeded replication fork progression in MAD2L2-depleted cells, in a manner that was more severe than for control cells (Fig. 2b,c). Taken together, our results point to a previously unappreciated role of MAD2L2 at stressed replication forks, that is different from its role in lesion bypass as a TLS factor.

### MAD2L2 is required for efficient replication fork restart and affects new origin firing

Although replication forks moved slower in the absence of MAD2L2, S-phase progression remained unaffected, as indicated by the comparable S-phase percentages of control and MAD2L2-depleted cells (Supplementary Fig. 1e,f). This could be reconciled by a more rapid restart of stalled forks and/or the firing of new replication origins upon replication stress. To test this hypothesis, we monitored fork restart by DNA fiber analysis in MAD2L2-depleted HeLa cells after release from HU treatment. We transiently labeled nascent DNA with CldU, treated the cells with 4 mM HU for 1 h, and then released cells in the presence of IdU. As there can be replication restart during HU treatment, we also analyzed fork restart with a different labeling scheme, keeping one of the nucleotide analogues (CldU) present with the drug. Both labeling schemes allow stalled forks (red only) to be distinguished from restarting forks (red-green) and allow identification of newly fired replication origins (green only). MAD2L2-depleted cells exhibited a severe defect in fork restart, as shown by the increased fork stalling and reduced percentage of restarting forks (Fig. 2d, Supplementary Fig. 2a). This suggests that, in the absence of MAD2L2, stalled forks restart slower and/or progress slower after their restart. Furthermore, new origins of replication were aberrantly fired (Fig. 2d, Supplementary Fig. 2a), indicating that MAD2L2 is required for efficient replication fork restart and is also involved in the regulation of dormant origin firing.

### MAD2L2 protects nascent DNA from degradation

To determine whether MAD2L2 plays a role in replication fork protection, we measured nascent DNA tract degradation using the DNA fiber technique. In this case, cells were treated with HU after sequential CldU and IdU labeling. When HU-stalled forks are not properly protected, the IdU-labeled nascent DNA at replication forks is degraded, leading to a reduced IdU:CldU ratio. In agreement with previous findings[8–10], we detected that BRCA1 is required for fork protection

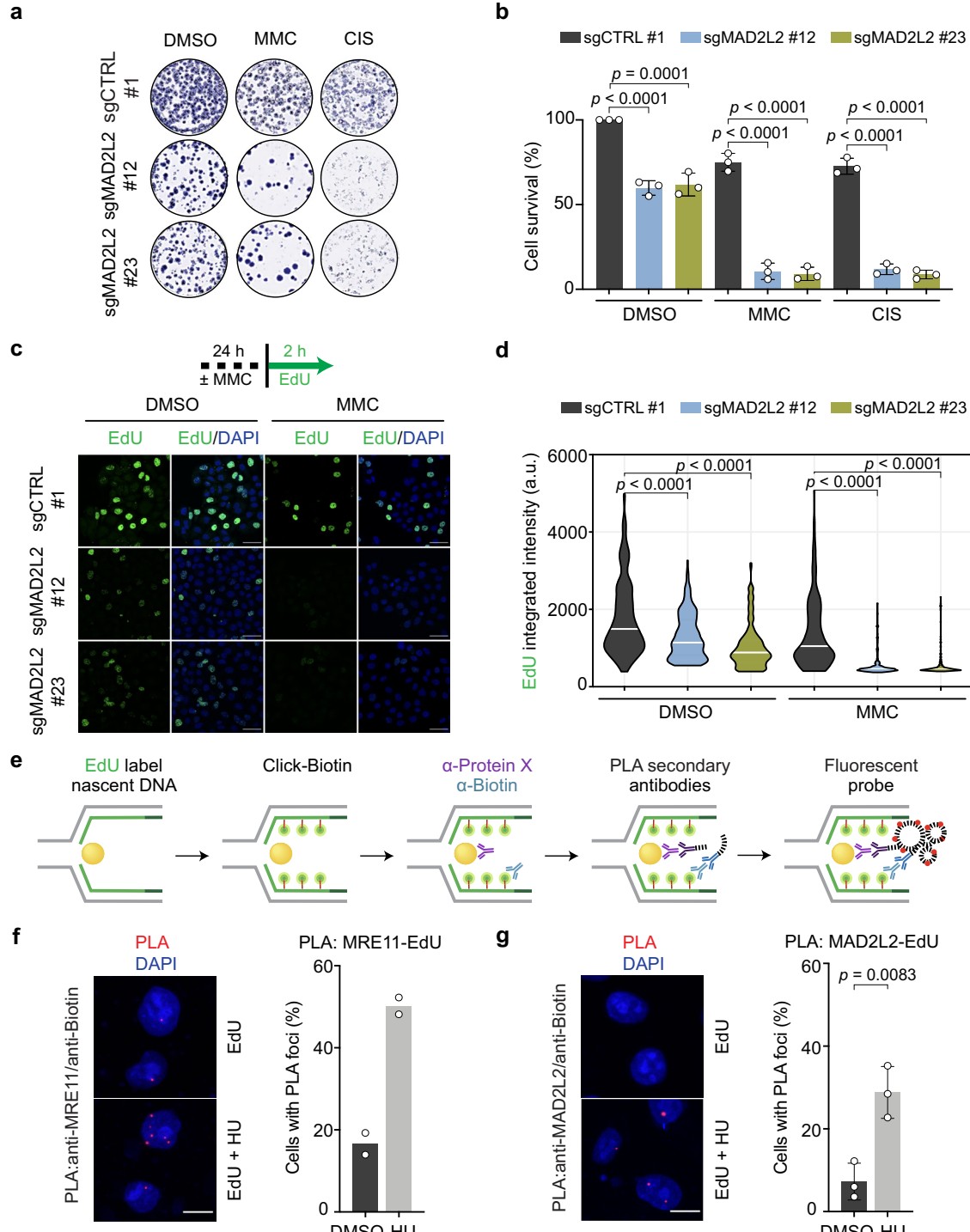

**Fig. 1 | MAD2L2 promotes global DNA replication during replicative stress.**
**a, b** Representative images (**a**) and quantification (**b**) of survival assays in control (sgCTRL) and MAD2L2-depleted HeLa cells, untreated or treated with 0.5 μM MMC or 1 μM CIS. Bars represent the mean ± SD. Each dot represents one of three independent experiments. Statistical analysis was performed according to one-way ANOVA with Dunnett's multiple comparisons test. **c, d** Assessment of DNA synthesis rates. Schematic with representative images (**c**) and quantification (**d**) of EdU incorporation intensity in control (sgCTRL) and MAD2L2-depleted HeLa cells. Cells were incubated with or without 5 μM MMC for 24 h, followed by EdU labeling for 2 h before harvesting. Scale bar, 50 μM. White bars represent the median of three

independent experiments. Statistical analysis was performed according to two-tailed Mann-Whitney test. **e** Schematic of the SIRF assay, which combines the proximity ligation assay (PLA) with EdU coupled click-iT chemistry to detect the association of target proteins to nascent DNA. Created with BioRender.com. **f, g** Representative images and quantification of the percentage of cells with MRE11/biotin PLA foci (**f**) or MAD2L2/biotin PLA foci (**g**) in HeLa control cells, untreated or treated with 4 mM HU for 2 h. Scale bar, 10 μM. Bars represent the mean ± SD. Each dot represents one of three independent experiments. Statistical analysis was performed according to unpaired two-tailed student t-test.

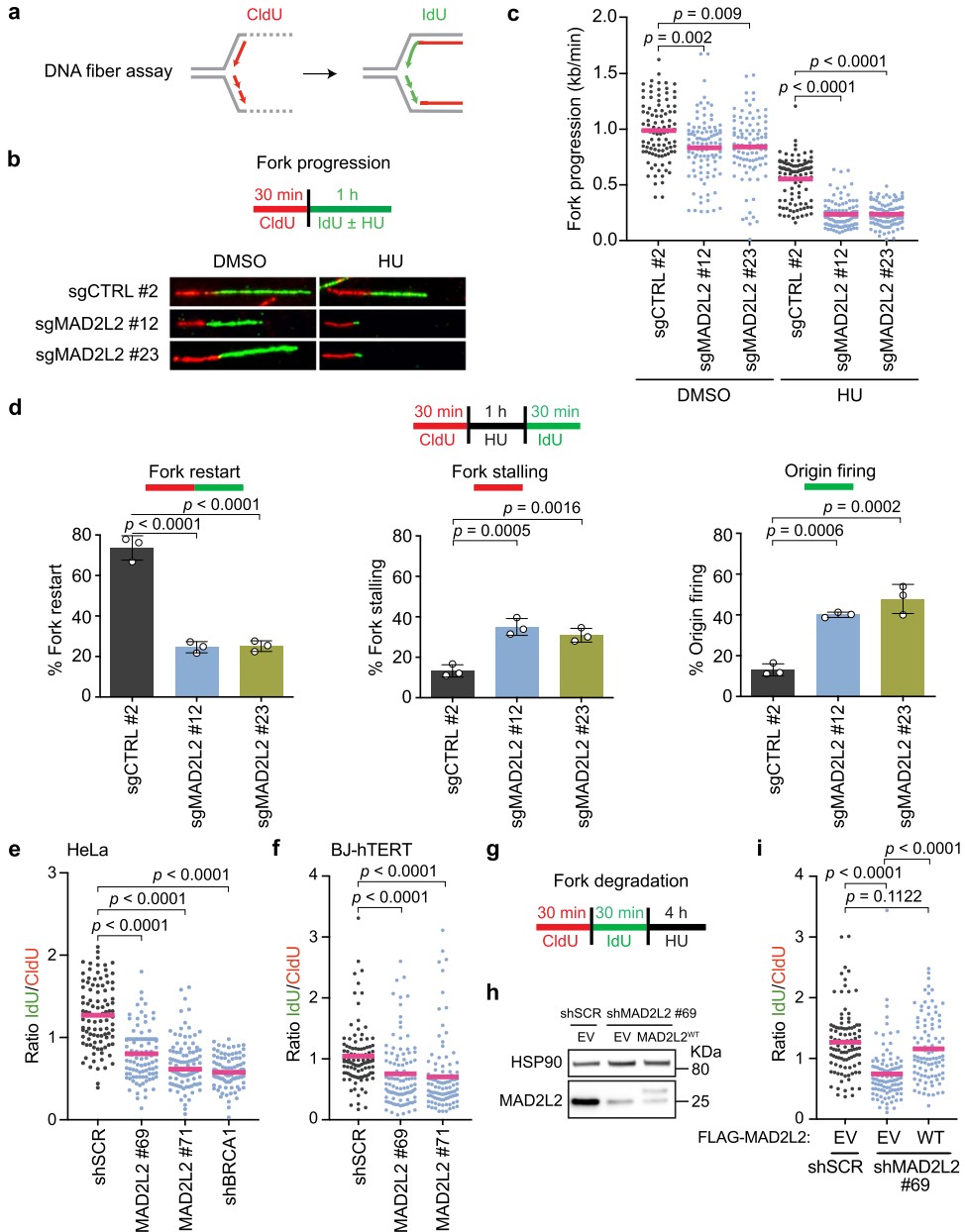

**Fig. 2 | MAD2L2 mediates protection and recovery of stalled replication forks.**
**a** Schematic of the DNA fiber assay, in which ongoing replication forks are sequentially labeled with two nucleotide analogues (CldU and IdU). **b** Top panel: schematic of fork progression assays. Cells were labeled with CldU (red) followed by treatment with or without 4 mM HU during the IdU (green) labeling period (1 h). Bottom panel: representative DNA fibers in control (sgCTRL) and MAD2L2-depleted HeLa cells. **c** Quantification of fork progression. Data was calculated by dividing tract lengths of IdU by the labeling time (kb/min). **d** Top panel: schematic of fork restart assays. Tract lengths of CldU and IdU were measured in control (sgCTRL) and MAD2L2-depleted HeLa cells upon HU release to assess fork restart, fork stalling and origin firing. Quantification is shown below. Bars represent the mean ± SD. Each dot represents one of three independent experiments. **e, f, g** Quantification (**e, f**) and schematic (**g**) of fork degradation assays. Cells were labeled with CldU followed by IdU and then treated with 4 mM HU to

induce fork stalling. The ratio of IdU to CldU tract length was quantified and plotted as readout for fork degradation in control (shSCR), MAD2L2-depleted, and BRCA1-depleted HeLa cells (**e**), and in control (shSCR) and MAD2L2-depleted BJ-hTERT cells (**f**). **h** Immunoblot analysis of MAD2L2-depleted Hela cells (shRNA #69) complemented with either empty vector (EV) or FLAG-tagged wild-type MAD2L2 (WT), as used in (**i**). HSP90 serves as loading control. A representative blot of three independent experiments is shown. **i** Fork degradation in MAD2L2-depleted HeLa cells is rescued after complementation with WT MAD2L2. Experimental conditions were similar as in (**e-g**). For the fiber assays in (**c, e, f, i**), a representative experiment of three independent replicates is shown. Pink bars in fiber plots represent the mean. Statistical analysis was performed according to two-tailed Mann-Whitney test. For the fiber assays in (**d**), statistical analysis was performed according to one-way ANOVA with Dunnett's multiple comparisons test. Additional replicates and combined fiber plots are provided in the Supplementary Information.

since the IdU:CldU ratios were reduced upon HU treatment of BRCA1-depleted cells (Fig. 2e, Supplementary Fig. 2b). Strikingly, HeLa cells depleted for MAD2L2 with multiple independent shRNAs or sgRNAs also displayed significantly reduced IdU:CldU ratios compared to control cells, indicating that newly synthesized DNA at stalled forks

gets degraded in absence of MAD2L2 (Fig. 2e, Supplementary Fig. 2c,d). These results were validated in nontransformed p53-deficient RPE1-hTERT *TP53-/-* cells and p53-proficient BJ-hTERT cells depleted for MAD2L2, indicating a general, noncell line specific phenotype (Fig. 2f,g, Supplementary Fig. 1c, Supplementary Fig. 2e,f). Of

note, we observed a cell-line-specific role for p53 in fork stabilization in RPE1-hTERT cells, as MAD2L2 depletion in p53-proficient RPE1-hTERT cells did not cause significant fork degradation (Supplementary Fig. 2g,h). While this may relate to previously reported roles of p53 in fork protection or remodeling of stalled forks[42], the mechanistic basis for this cell-line specificity of p53 is unclear at this point.

MAD2L2 depletion also resulted in nascent DNA degradation upon treatment with 300 μM HU, a low HU dose reported to cause replication fork stalling but not replication fork collapse (Supplementary Fig. 2i)[43,44]. While we cannot formally exclude the formation of a few DSBs with the low dose of HU, these results are supportive of MAD2L2 acting at stalled forks rather than at broken forks. Finally, to further verify that the effects of MAD2L2 knockdown on replication forks are specifically due to loss of MAD2L2 function and reflect a specific activity of MAD2L2, we depleted endogenous MAD2L2 with an shRNA targeting the MAD2L2 3' UTR and restored MAD2L2 expression with exogenous FLAG-tagged MAD2L2. Indeed, exogenous MAD2L2 restored replication fork stability in MAD2L2-depleted HeLa cells, confirming that loss of MAD2L2 function is responsible for the decreased fork protection in these cells (Fig. 2h,i). Taken together, our results indicate that MAD2L2 is an essential factor in the protection of nascent DNA.

### The MRE11 nuclease drives replication fork degradation in MAD2L2-deficient cells

Previous studies have shown that the nascent DNA degradation observed upon HU treatment in multiple contexts, including BRCA1/2-deficient cells[8,9], depends on the formation of reversed replication forks. To determine if this is also the case in MAD2L2-depleted cells, we used small interfering RNAs (siRNAs) to deplete selected factors involved in fork reversal, including the recombinase RAD51[3,4] and the motor proteins SMARCAL1 and FBH1[1] (Supplementary Fig. 3a-d). We found that near complete inhibition of RAD51 and SMARCAL1, but not FBH1, suppresses fork degradation in MAD2L2-depleted cells (Fig. 3a). These results indicate that RAD51 and SMARCAL1 generate a DNA structure, namely a reversed fork, that is degraded when MAD2L2 is inactivated.

Reversed replication forks have exposed single-stranded DNA (ssDNA) or double-stranded DNA (dsDNA) ends that can serve as a substrate for nucleases such as MRE11, CtIP, DNA2 and EXO1[5-7]. Given that MRE11 initiates the resection of nascent DNA strands[45], we hypothesized that MRE11 might mediate reversed fork degradation upon MAD2L2 depletion. To address this, we treated MAD2L2-depleted HeLa cells with the MRE11 inhibitor mirin or an siRNA against MRE11 (Fig. 3b,c, Supplementary Fig. 3e,f). Indeed, nascent DNA degradation in the absence of MAD2L2 was rescued by both mirin and siRNA-mediated MRE11 depletion, indicating that MRE11 is the nuclease responsible for the observed fork degradation. Notably, mirin did not rescue the reduced EdU incorporation rates of MAD2L2-depleted cells (Supplementary Fig. 3g), suggesting that the functions of MAD2L2 in fork protection and fork restart are uncoupled.

Since excessive DNA fork degradation has been linked to increased genome instability[2,9], we examined metaphase spreads of control and MAD2L2-depleted cells after treatment with HU. Untreated MAD2L2-depleted cells did not show a significant increase in aberrant chromosomes. At 24 h after release from HU treatment, however, MAD2L2-depleted cells displayed elevated chromosomal instability, in particular chromosome and chromatid breaks (Fig. 3d,e). In response to 4 h HU treatment, MAD2L2-depleted cells also displayed increased phosphorylation of H2AX at serine 139 (γ-H2AX), a marker of both DSBs and extensive ssDNA at stalled forks[46] (Fig. 3f). Neutral comet assays did not reveal an increase in DSBs after a 4 h HU treatment, nor were any significant changes in DSBs detected between control and MAD2L2-depleted cells (Fig. 3g,h). Thus, we hypothesize that the increase in chromosomal aberrations in MAD2L2-depleted cells

originated from those replication forks that stalled/collapsed after release from HU. Taken together, our results indicate that MAD2L2 contributes to genome maintenance by preventing excessive processing of stalled replication forks by MRE11.

### MAD2L2 acts at stalled replication forks independently of the shieldin complex

MAD2L2 has several binding partners, which could potentially mediate its fork protection activity, including SHLD1, SHLD2 and SHLD3 that form the shieldin complex with MAD2L2. Given the important activity of shieldin in controlling nuclease-mediated resection at DSBs, and the implication of multiple upstream and downstream factors of shieldin in replication fork protection, we considered shieldin to be a prime candidate for mediating the effects of MAD2L2 at stalled replication forks. To examine the activity of shieldin at replication forks, we first generated stable polyclonal knockout HeLa cells for SHLD1, SHLD2 and SHLD3 using previously validated lentiviral sgRNAs[19]. Besides verification of gene editing by TIDE[47] and/or qPCR (Supplementary Data 3, Supplementary Fig. 4a), we validated the shieldin deficiency in the knockout cell lines by assessing loss of shieldin-mediated end resection inhibition. As expected from previous work[19,22-25], loss of each shieldin subunit led to increased single-stranded DNA (ssDNA) accumulation upon (IR)-induced DNA damage (Fig. 4a-c), reflecting elevated DNA resection and thus confirming the loss of shieldin activity in these cells.

We then addressed the consequences of SHLD1, SHLD2 and SHLD3 deficiency on replication fork protection. Surprisingly, depletion of either of these shieldin subunits did not affect fork stability upon HU treatment, as shown by the similar IdU:CldU ratios between knockout and control cells (Fig. 4d). In line with this, additional knockdown of SHLD2 in MAD2L2-depleted HeLa cells also had no effect on the level of fork degradation associated with MAD2L2 loss (Supplementary Fig. 4 b,c). This suggests that while shieldin protects DSB ends from excessive resection, it does not appear to have this protective function at HU-stalled replication forks. Moreover, it indicates that MAD2L2 promotes stalled fork stabilization in a process that is independent of shieldin.

We further verified this pathway differentiation by addressing epistasis in fork protection between MAD2L2 and 53BP1/RIF1, the upstream recruiters of shieldin. In line with previous reports[28,32-34], depletion of 53BP1 and RIF1 individually causes nascent DNA degradation upon HU treatment (Fig. 4e, Supplementary Fig. 4d-f). However, whereas MAD2L2 depletion resulted in increased fork degradation rates upon HU in control and 53BP1-depleted cells, MAD2L2 depletion did not further increase fork degradation rates upon HU in RIF1-depleted cells (Fig. 4e, Supplementary Fig. 4d-f). This indicates that MAD2L2 acts in fork protection in a different pathway than 53BP1, but has an epistatic relationship with RIF1 in fork protection. Interestingly, 53BP1 was previously shown to protect forks remodelled by FBH1, while we found FBH1 to be dispensable for fork degradation in MAD2L2-depleted cells (Fig. 3a). It is then likely that 53BP1 and MAD2L2 display substrate preferences at stalled replication forks and thus act at distinct steps during fork protection, to individually block nucleolytic activities.

Lastly, since both 53BP1 and RIF1 suppress the activity of the DNA2 nuclease at stalled forks[28,32-34], we also examined whether MAD2L2 blocks DNA2-mediated fork degradation. As shown in Fig. 4f and Supplementary Fig. 4g, siRNA-mediated depletion of DNA2 did not alleviate the uncontrolled fork degradation observed in MAD2L2-depleted cells. Thus, MAD2L2 appears to act independently of 53BP1 and RIF1, and selectively counteracts MRE11-mediated degradation. These latter results seem to contradict the epistatic relationship we observed between MAD2L2 and RIF1 in fork protection (Fig. 4e). Although the underlying reason is unclear, we hypothesize that stalled forks processed by MRE11 in absence of MAD2L2 may be prohibited from further processing by other nucleases such as DNA2, or may not

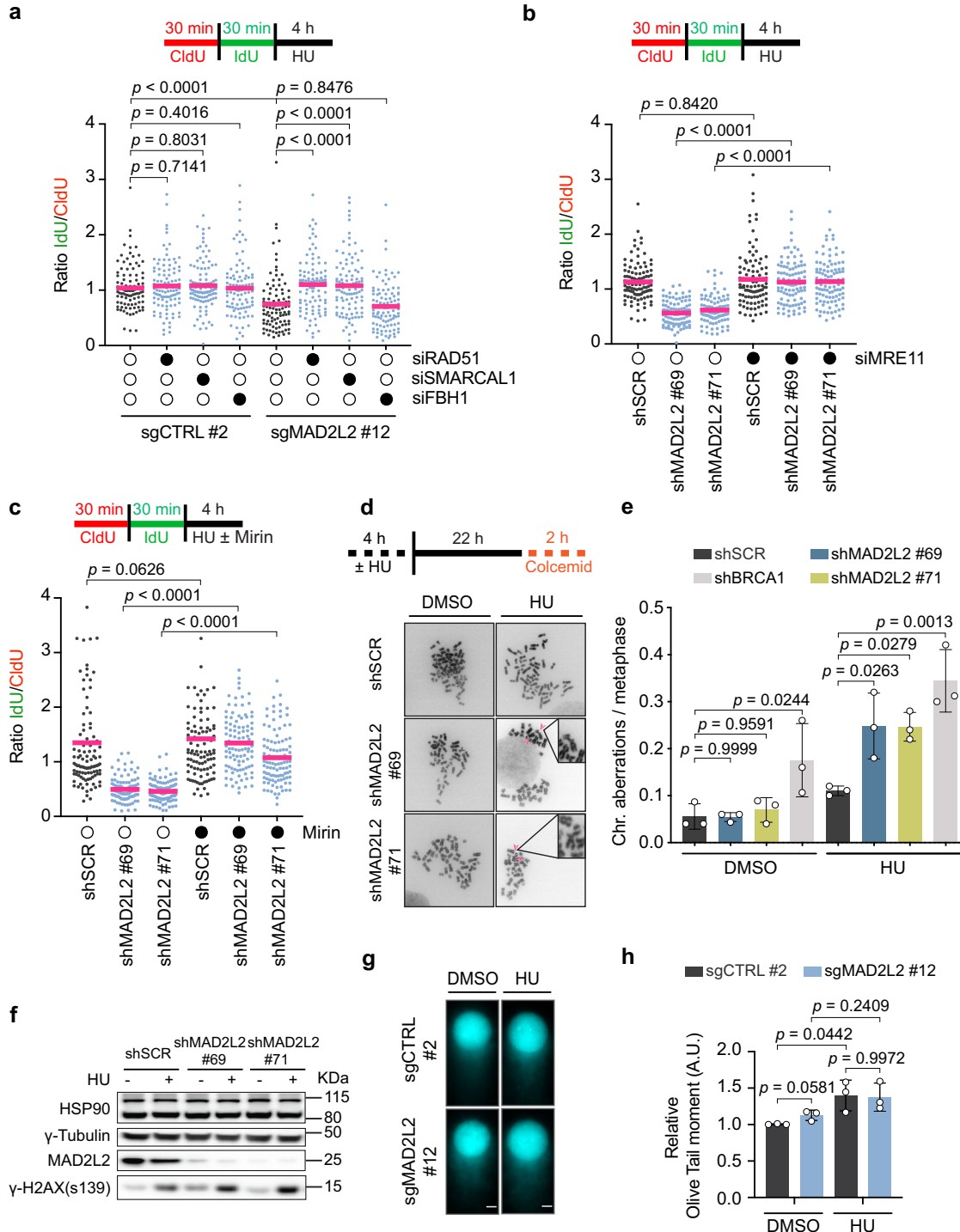

**Fig. 3 | The MRE11 nuclease drives reversed fork degradation in MAD2L2-deficient cells. a** Schematic and quantification of fork degradation assays in control (sgCTRL) and MAD2L2-depleted HeLa cells, transfected with an siRNA control (open circles) or the indicated siRNAs (closed circles). Cells were incubated in 4 mM HU for 4 h before being harvested. A representative experiment of three independent replicates is shown. **b** Control (shSCR) and MAD2L2-depleted HeLa cells were transfected with an siRNA control or an siRNA targeting MRE11. Cells were treated and labeled as indicated above. A representative experiment of three independent replicates is shown. **c** Control (shSCR) and MAD2L2-depleted HeLa cells were labeled as indicated above and subsequently treated with 4 mM HU, with or without the MRE11 inhibitor mirin (50 μM) for 4 h. A representative fiber experiment from three independent biological replicates is shown. **d** Schematic and representative images of metaphase spreads in control (shSCR), MAD2L2-depleted, and BRCA1-depleted HeLa cells following treatment with 4 mM HU for 4 h and recovery for 24 h. Arrows indicate chromosome/chromatid breaks. Scale bar, 10 μM. **e** Quantification

of chromosomal aberrations at 24 h after release from a 4 h treatment with 4 mM HU. Bars represent the mean ± SD. Each dot represents one of three independent experiments. **f** Immunoblot analysis of γ-H2AX levels in control and MAD2L2-depleted HeLa cells, untreated or treated with 4 mM HU for 4 h. HSP90 and γ-Tubulin serve as loading controls. Representative blots of two independent experiments are shown. **g, h** Representative images (**g**) and quantification (**h**) of the neutral comet assay in control (sgCTRL) and MAD2L2-depleted HeLa cells following treatment with 4 mM HU for 4 h. Scale bar, 10 μM. Statistical analysis for the fiber assays in (**a-c**) was performed according to two-tailed Mann-Whitney test. Pink bars represent the mean. Additional replicates and combined fiber plots are provided in the Supplementary Information. Statistical analysis in (**e**) was performed according to one-way ANOVA with Dunnett's multiple comparisons test. Statistical analysis in (**h**) was performed according to two-way ANOVA with Tukey's multiple comparisons test.

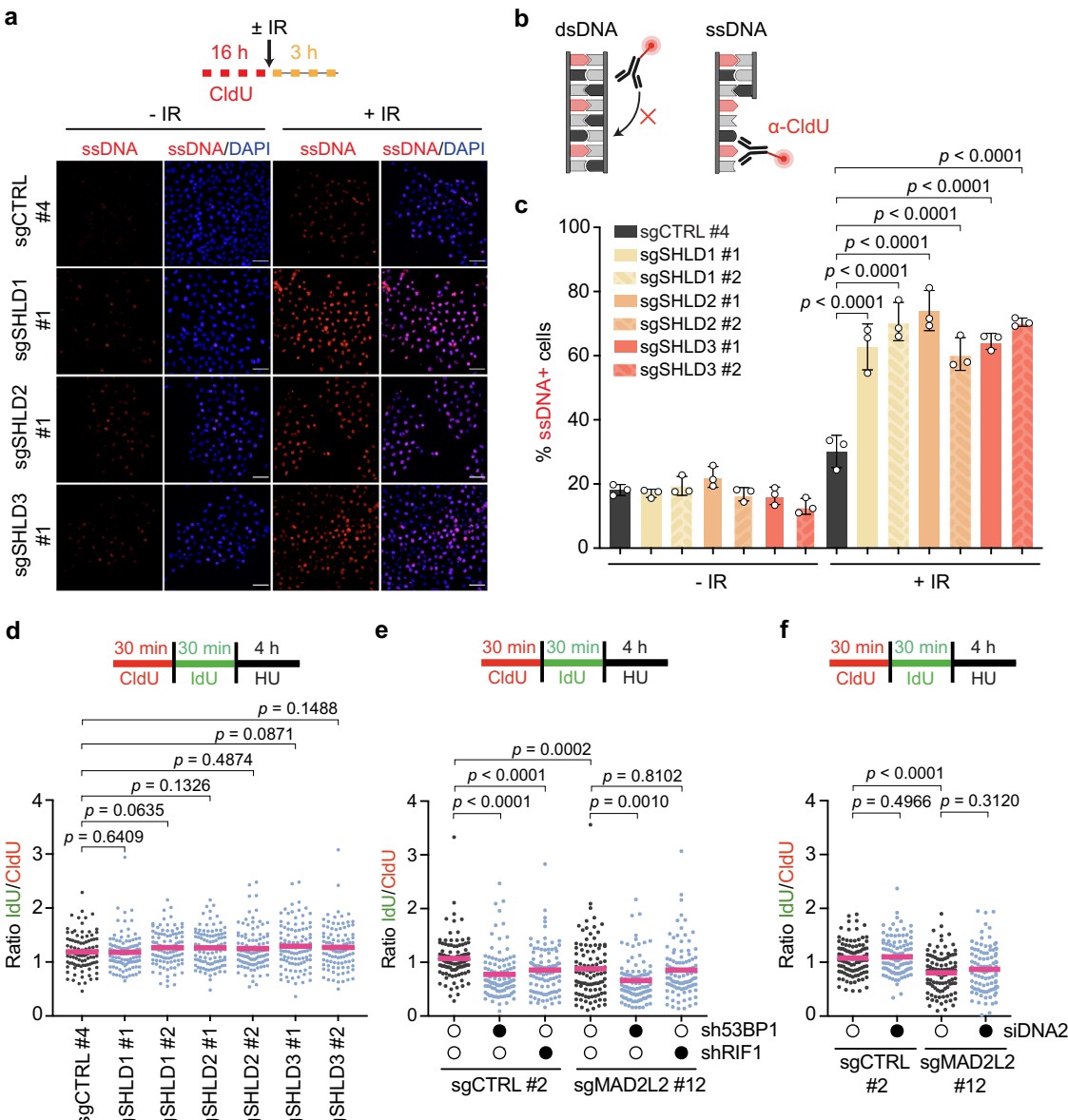

**Fig. 4 | MAD2L2 acts at stalled replication forks in a shieldin-independent manner. a** Schematic and representative images of ssDNA-positive cells in control (sgCTRL) and SHLD1-, SHLD2- or SHLD3-depleted HeLa cells. Following initial labeling with CldU (red) for 16 h, cells were either left untreated or harvested 3 h after irradiation (IR) with 25 Gy. Scale bar, 50 μM. Quantification of three independent experiments is shown in (**c**). **b** Schematic of the native CldU assay to detect ssDNA. Created with BioRender.com. **c** Quantification of ssDNA-positive cells. Bars represent the mean ± SD. Each dot represents one of three independent experiments. Statistical analysis was performed according to one-way ANOVA with Dunnett's multiple comparisons test. **d** Schematic and quantification of fork degradation assays in control and SHLD1-, SHLD2- or SHLD3-depleted HeLa cells. Cells were treated with 4 mM HU for 4 h before being harvested. A representative

fiber experiment from three independent biological replicates is shown. **e** Schematic and quantification of fork degradation assays in control and MAD2L2-depleted HeLa cells transduced with the indicated shRNAs. Experimental conditions were similar as in (**d**). Open circles, control shRNA; closed circles, 53BP1 or RIF1 shRNA. A representative fiber experiment from three independent biological replicates is shown. **f** Control and MAD2L2-depleted HeLa cells were transfected with siRNA control luciferase or siRNA targeting DNA2. Cells were treated and labeled as indicated above. A representative fiber experiment from two independent biological replicates is shown. Statistical analysis for the fiber assays in (**d**–**f**) was performed according to two-tailed Mann-Whitney test. Pink bars represent the mean. Additional replicates and combined fiber plots are provided in the Supplementary Information.

be substrates of RIF1-mediated protection, causing additional depletion of RIF1 to be without effect in MAD2L2-depleted cells. All together, our data highlight an unified molecular mechanism for repair-independent functions of MAD2L, 53BP1, and RIF1 in replication fork stabilization.

## MAD2L2 and REV3L cooperate in safeguarding replication fork integrity

Given that MAD2L2 also acts in a complex (Polζ) with REV3L, we sought to determine whether the protection of nascent DNA by MAD2L2

would be dependent on this interaction. We first established doxycycline-inducible REV3L knockout HeLa cells and examined the cellular response to different replication perturbants. Similar to the loss of MAD2L2, and in line with previously reported findings[48,49], REV3L depletion caused hypersensitivity to DNA crosslinking agents (Fig. 5a, Supplementary Fig. 5a,b) and resulted in decreased replication rates after MMC treatment (Fig. 5b). However, REV3L-depleted cells also showed compromised replication dynamics in unchallenged conditions (Fig. 5b,c), which were further worsened upon HU treatment (Fig. 5c), confirming an earlier observation[50]. Yet, flow cytometry

analysis of DNA content in asynchronous REV3L-depleted and control cells showed no specific cell cycle arrest point (Supplementary Fig. 5c,d), suggesting that REV3L loss has no significant effect on S-phase progression. Taken together, our data and that of others indicate that REV3L has an important function at the progressing replication fork, both during physiological and exogenous replication stress.

We next addressed the role of REV3L in fork protection by DNA fiber assays. Interestingly, REV3L-depleted cells displayed significantly reduced IdU:CldU ratios compared to control cells (Fig. 5d), indicating that REV3L also protects replication forks against nucleolytic degradation. Treatment of REV3L-depleted cells with mirin restored normal replication fork stability (Fig. 5d) to a similar extent as observed in MAD2L2-depleted cells (Fig. 3c, Supplementary Fig. 3f). Additionally, we detected that in the absence of REV3L stalled forks exhibit defective fork restart and an increase in origin firing after HU treatment (Fig. 5e), as also observed for MADL2-depleted cells (Fig. 2d, Supplementary Fig. 2a). Importantly, the additional knockdown of MAD2L2 in REV3L-depleted cells did not further increase the level of fork degradation (Fig. 6a, Supplementary Fig. 6a,b). This indicates that MAD2L2 is epistatic with REV3L and that both act in the same mechanism to facilitate replication fork protection. In line with this epistatic relationship, and suggesting that MAD2L2 mediates the recruitment of REV3L to stalled forks, SIRF assays detected increased association of endogenous REV3L to nascent DNA after HU treatment in control cells, but not in MAD2L2-depleted cells (Fig. 5f).

Lastly, we hypothesized that the observed defects in fork restart and/or excessive degradation rates could result in the exposure of ssDNA, which can be a source of genome instability[51]. To study this, we treated CldU-labeled cells with 4 mM HU for 2 h, a condition that induces severe fork stalling and results in the accumulation of ssDNA[52]. Subsequently, we probed for CldU under non-denaturing conditions to quantify the levels of ssDNA present. As shown in Fig. 6b and Supplementary Fig. 6c, MAD2L2 and REV3L depletion led to widespread ssDNA accumulation. This indicates that MAD2L2 and REV3L promote replication during stress and prevent aberrant genome-wide ssDNA formation.

### MAD2L2 activity at stressed replication forks requires REV1 and the polymerase activity of REV3L

Upon replication fork stalling at DNA damage sites, REV1 recruits Polζ through MAD2L2 to promote TLS[13]. This functional link between REV1 and Polζ prompted us to investigate whether REV1 is also required for MAD2L2/REV3L-mediated fork protection. To test this possibility, we established doxycycline-inducible REV1 knockout HeLa cells (Supplementary Fig. 5d). We first confirmed that REV1-depleted cells are more sensitive to DNA crosslinking agents than control cells (Fig. 6c, Supplementary Fig. 6e) and that treatment with MMC leads to reduced replication rates (Fig. 6d)[37]. REV1 depletion was not associated with defects in cell cycle progression or cell survival, nor with impaired replication dynamics under unchallenged conditions (Fig. 6c-e, Supplementary Fig. 6f,g). Thus, unlike MAD2L2/REV3L, REV1 appears dispensable for DNA synthesis under physiological conditions. However, in line with recent work on REV1 inhibitors[53,54], REV1-depleted cells displayed reduced replication fork progression upon treatment with HU (Fig. 6e). Moreover, following HU treatment, REV1-depleted cells also exhibited defective fork restart (Fig. 6g) and, consistent with earlier work[55], a significant increase in MRE11-dependent fork degradation (Fig. 6f). Importantly, knockdown of MAD2L2 in REV1-depleted cells resulted in no additional impairment of fork protection, demonstrating that these two factors are epistatic for replication fork protection (Fig. 6h, Supplementary Fig. 6h,i).

A key question we asked is whether REV1 and REV3L require their DNA polymerase activities for replication fork protection. To examine this, we expressed full-length wild-type REV1[56] and catalytic inactive

mutant REV1, in which the active site residues D570 and E571 have both been changed to Alanines. Interestingly, both wild-type and D570A/E571A catalytic inactive mutant REV1 rescued the fork protection defect in REV1-depleted cells (Fig. 7a, Supplementary Fig. 7a), indicating that REV1's polymerase activity is not required for fork protection. REV1 may thus play a more structural role in fork protection, recruiting and coordinating MAD2L2/REV3L at stalled replication forks. In contrast, defective fork protection in REV3L-depleted cells was reverted by complementation with wild-type REV3L, but not catalytic inactive mutant (D2781A/D2783A) REV3L (Fig. 7b, Supplementary Fig. 7b)[57]. Thus, REV3L polymerase-dependent DNA synthesis is essential for replication fork protection under conditions of nucleotide shortage.

Finally, we addressed whether this collaborative role of MAD2L2/REV3L/REV1 prevents replication stress-associated genomic instability. Loss of REV3L already by itself led to increased spontaneous chromosomal aberrations (Fig. 7c, Supplementary Fig. 7c). This is consistent with previous data and with REV3L being essential, and has been ascribed at least in part to a critical role for REV3L in maintaining common fragile site stability[58]. Chromosomal aberrations also increased when REV3L- and MAD2L2-depleted cells, but not REV1-depleted cells, were challenged with HU-induced replication stress, and this effect was greatly exacerbated when depletion of REV3L was combined with MAD2L2 loss (Fig. 7c, Supplementary Fig. 7c). Moreover, REV3L and MAD2L2 double-depleted cells showed an increase in complex chromosomal aberrations (radial figures), suggestive of aberrant processing of DSBs. We hypothesize that the particularly high prevalence of chromosomal aberrations in cells double-depleted for REV3L and MAD2L2 and treated with HU is likely reflecting both the role of REV3L in maintaining common fragile sites[58], which represent 'hot spots' of chromosomal breakage, and the role of MAD2L2 in DSB repair. The absence of increased chromosomal aberrations in REV1-depleted cells suggests that the genomic loci confronted with increased fork degradation and ssDNA generation upon HU are mostly restored correctly in these cells, while this more regularly fails in REV3L- and MAD2L2-depleted cells.

## Discussion

DNA replication stress is a major driving force of genome instability during tumorigenesis and is exploited in cancer therapy[59]. Several proteins commonly associated with DSB repair have been implicated in replication fork stabilization[8-10,28,32,35], thus contributing to genome stability in unanticipated ways. Herein, we report that the NHEJ promoting factor MAD2L2 prevents uncontrolled nascent DNA degradation by MRE11 at stalled replication forks and facilitates replication restart after stress. Surprisingly, while this activity resembles the suppression of excessive DNA resection at DSBs by MAD2L2 within shieldin, this function of MAD2L2 at replication forks is not dependent on its interaction with the shieldin subunits, since fork protection remained unaffected by shieldin loss. Instead, we find that MAD2L2 acts with the TLS components, REV3L and REV1, to stabilize replication forks and limit the accumulation of ssDNA, thereby alleviating DNA replication stress.

Of particular interest, our data indicate that the function of MAD2L2 and REV3L in the DNA replication stress response is not limited to exogenous stress, as EdU incorporation and replication fork progression were also reduced in unperturbed conditions upon inactivation of MAD2L2 and REV3L (Figs. 1d, 2c, 5b, c). Certain genomic regions are known to be more challenging to replicate than others due to repetitive sequences, DNA secondary structures and frequent collisions of the replication fork with RNA polymerases in areas of high transcriptional activity[60]. It is thus conceivable that at any one time a considerable portion of replication forks stalls at these regions and requires active MAD2L2/REV3L to drive fork progression or restart. In support of this, cells were recently found to be highly dependent on

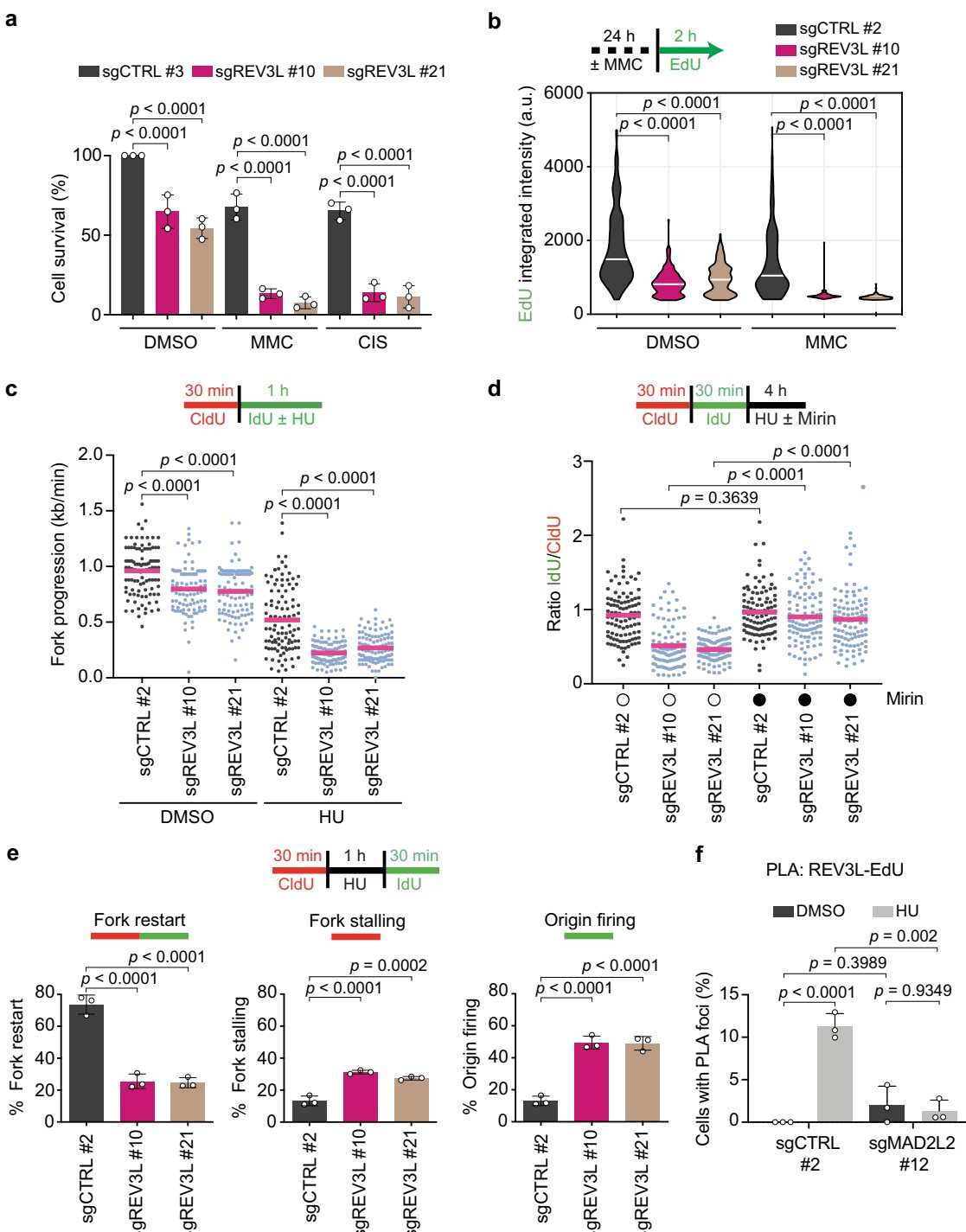

**Fig. 5 | REV3L is essential to protect and restart stalled replication forks.**
**a** Quantification of survival assays in control (sgCTRL) and REV3L-depleted HeLa cells, untreated or treated with 0.5 µM MMC or 1 µM CIS. Bars represent the mean ± SD. Each dot represents one of three independent experiments. **b** Schematic and quantification of EdU incorporation intensity in control and REV3L-depleted HeLa cells. Cells were incubated with or without 5 µM MMC for 24 h, followed by EdU labeling for 2 h before harvesting. White bars represent the median of three independent experiments. **c** Schematic and quantification of fork progression assays in control and REV3L-depleted HeLa cells, untreated or treated with 4 mM HU for 1 h. A representative fiber experiment from two independent biological replicates is shown. **d** Schematic and quantification of fork degradation assays. Control and REV3L-depleted HeLa cells were treated with 4 mM HU ± 50 µM mirin for 4 h. Open circles, no treatment; closed circles, treatment. A representative fiber experiment from three independent biological replicates is shown.

**e** Schematic and quantification of fork restart, fork stalling, and origin firing in control and REV3L-depleted HeLa cells treated with 4 mM HU for 1 h as indicated. Data are represented as mean ± SD. Each dot represents one of three independent experiments. **f** Quantification of the percentage of cells with REV3L/biotin PLA foci in control and MAD2L2-depleted HeLa cells, untreated or treated with 4 mM HU for 2 h. Bars represent the mean ± SD. Each dot represents one of three independent experiments. Statistical analysis in (**a, e**) was performed according to one-way ANOVA with Dunnett's multiple comparisons test. Statistical analysis in (**b–d**) was performed according to two-tailed Mann-Whitney test. Pink bars in the fiber plots represent the mean. Additional replicates and combined fiber plots are provided in the Supplementary Information. Statistical analysis in (**f**) was performed according to two-way ANOVA with Tukey's multiple comparisons test.

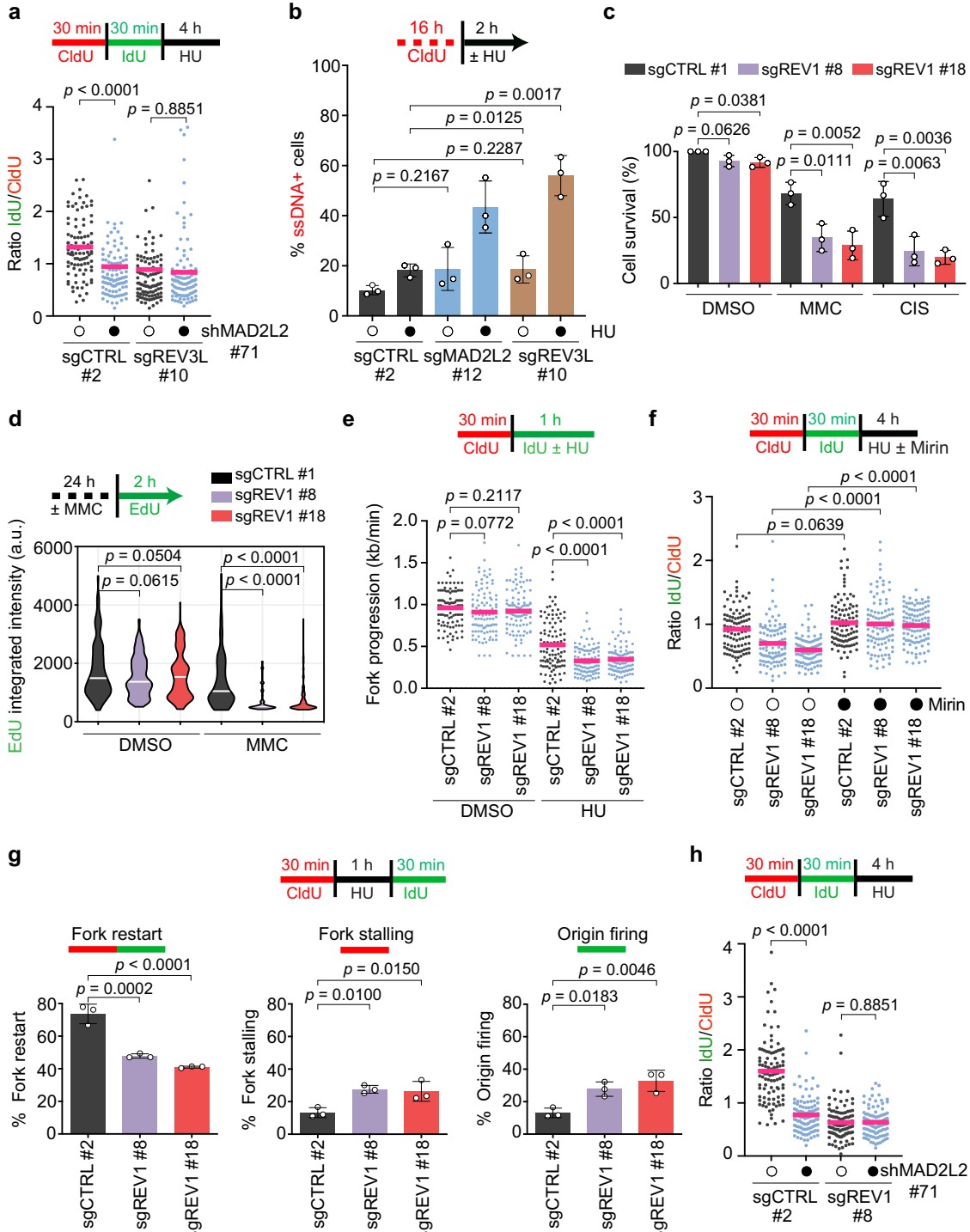

**Fig. 6 | MAD2L2 activity at stressed replication forks prevents ssDNA accumulation and requires REV3L and REV1. a** Schematic and quantification of fork degradation assays in control (sgCTRL) and REV3L-depleted HeLa cells, transduced with the indicated shRNAs and treated with 4 mM HU for 4 h. **b** Schematic and quantification of ssDNA-positive cells after CldU labeling for 16 h, followed by mock treatment (open circles) or treatment with 4 mM HU for 2 h (closed circles). Bars represent the mean ± SD of three independent experiments. **c** Quantification of survival assays in control and REV1-depleted HeLa cells, untreated or treated with 0.5 μM MMC or 1 μM CIS. Each dot represents one of three independent experiments. **d** Schematic and quantification of EdU incorporation intensity in control and REV1-depleted HeLa cells, untreated or treated with 5 μM MMC for 24 h, followed by EdU labeling for 2 h. White bars represent the median of three independent experiments. **e** Schematic and quantification of fork progression assays in control and REV1-depleted HeLa cells, untreated or treated with 4 mM HU for 4 h. **f** Schematic and quantification of fork degradation assays in control and REV1-depleted HeLa cells, treated with 4 mM HU ± 50 μM mirin for 4 h. Open circles, no treatment; closed circles, treatment. **g** Schematic and quantification of fork restart assays in control and REV1-depleted HeLa cells, treated with 4 mM HU for 1 h as indicated. Bars represent the mean ± SD of three independent experiments. **h** Schematic and quantification of fork degradation assays in control and REV1-depleted HeLa cells transduced with control shRNA (shSCR, open circles) or MAD2L2 shRNA (closed circles). Experimental conditions were similar as in (**a**). Pink bars in fiber plots represent the mean. Statistical analysis in (**a, d, e, f, h**) was performed according to two-tailed Mann-Whitney test. A representative fiber experiment from two (**h**) or three (**a, d, e, f**) independent biological replicates is shown. Additional replicates and combined fiber plots are provided in the Supplementary Information. Statistical analysis in (**b, c, g**) was performed according to one-way ANOVA with Dunnett's multiple comparisons test.

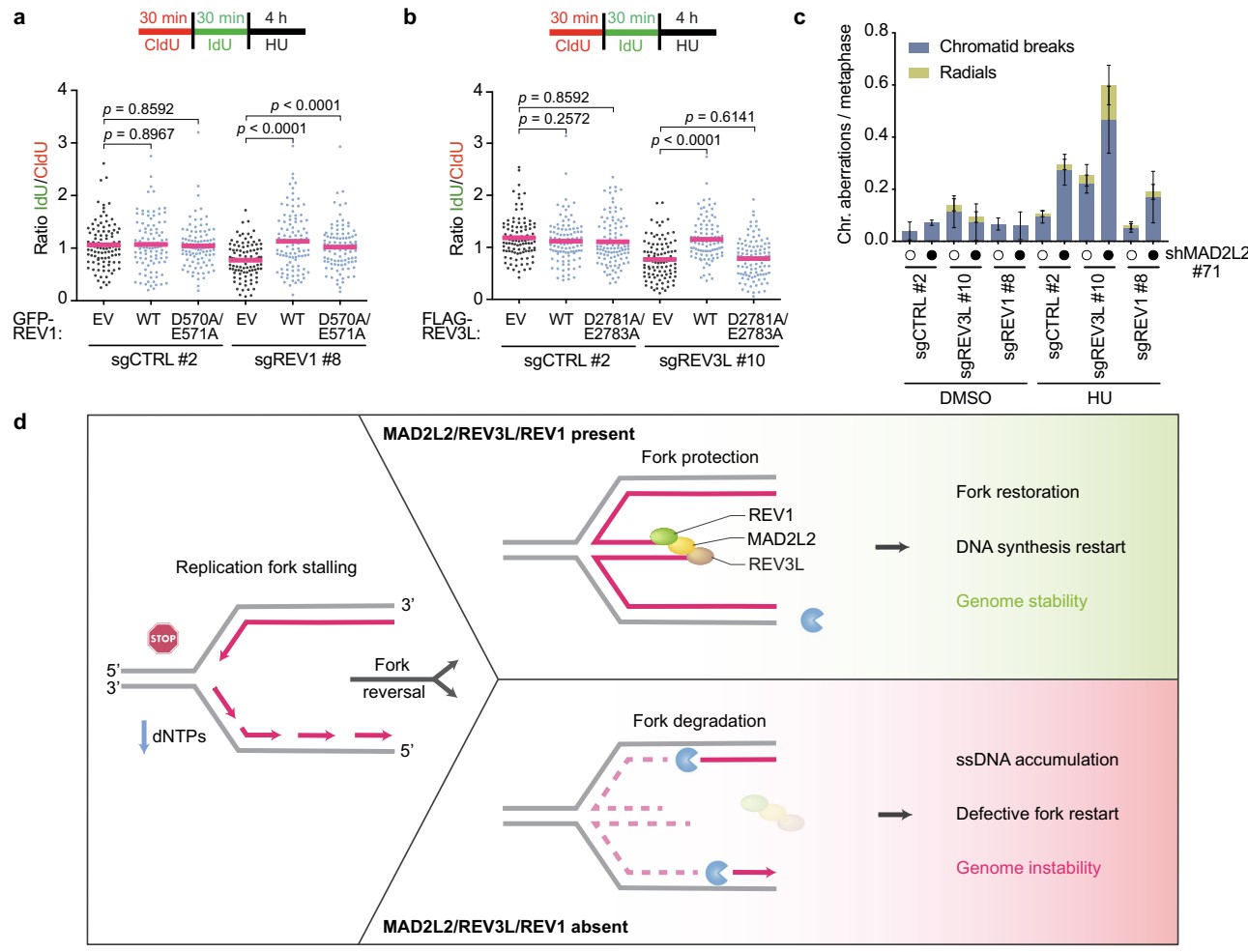

**Fig. 7 | MAD2L2, REV1 and REV3L polymerase activity are required for replication fork protection and genome maintenance. a** Schematic and quantification of fork degradation assays in control (sgCTRL) and REV1-depleted HeLa cells complemented with either empty vector (EV), GFP-tagged wild-type REV1 (WT), or GFP-tagged catalytic inactive mutant REV1 (D570A/E571A). A representative fiber experiment from three independent biological replicates is shown. **b** Schematic and quantification of fork degradation assays in control and REV3L-depleted HeLa cells complemented with either empty vector (EV), FLAG-tagged wild-type REV3L (WT), or FLAG-tagged catalytic inactive mutant REV3L (D2781A/E2783A). A representative fiber experiment from three independent biological replicates is shown. Pink bars in fiber plots represent the mean. Statistical analysis in (**a**, **b**) was performed according to two-tailed Mann-Whitney test. Additional replicates and combined fiber plots are provided in the Supplementary Information. **c** Quantification of chromosomal aberrations in control, REV1- and REV3L-depleted HeLa cells transduced with a control shRNA (shSCR, open circles) or MAD2L2 shRNA (closed circles). Cells were harvested following treatment with 4 mM HU for 4 h and recovery for 24 h. Bars represent the mean ± SD of three independent biological replicates. **d** Schematic model depicting the role of MAD2L2/REV3L/REV1 in fork protection and genome maintenance. Upon replication fork stalling due to e.g. nucleotide depletion, remodeling of stalled forks can generate a regressed, four-stranded DNA structure that is susceptible to the action of cellular nucleases such as MRE11. MAD2L2, in collaboration with REV1 and REV3L, stabilizes and protects reversed replication forks, thereby allowing efficient restart of DNA synthesis. In the absence of MAD2L2/REV3L/REV1, reversed forks undergo extensive MRE11-dependent nucleolytic degradation, which compromises fork restart and results in ssDNA accumulation and genome instability. Created with BioRender.com.

MAD2L2 and REV3L to cope with transcription-replication conflicts[61,62]. However, whether MAD2L2/REV3L replace (or coexist with) replicative polymerases within the replisome in this context is still an open question. Interestingly, even though MAD2L2/REV3L and REV1 work closely together in TLS, and MAD2L2-mediated protection of HU-stalled replication forks depends on REV1 (Fig. 6h), REV1 depletion did not affect replication fork progression under unperturbed conditions, whereas MAD2L2 and REV3L depletion did (Figs. 1d, 2c, 5b, c, Fig. 6d,e). This suggests that the recruitment of MAD2L2/REV3L in the context of endogenous replication stress is independent of REV1 and relies on another factor or mechanism.

Our study shows that MAD2L2 protects cells against multiple replication poisons, such as CIS, MMC[13], HU and aphidicolin. While the role of MAD2L2 in the replication bypass of DNA lesions as noncatalytic component of Polζ is well described[15], its role in protecting stalled forks

from degradation was thus far unknown. Given that Polζ lacks proofreading activity and is hence potentially more mutagenic than normal replicative polymerases[63], its implication in fork protection is intriguing. For instance, sustained Polζ activity at the fork could lead to the introduction of DNA base-pair mismatches and result in fork destabilization. However, the lack of involvement of REV1 catalytic activity in fork protection may limit this risk in comparison to the risk of mutagenesis associated with Polζ activity in TLS (Fig. 7a). In TLS, REV1 functions as an inserter polymerase that incorporates deoxycytidine (dC) opposite to a DNA lesion, whether being correct or incorrect. Polζ efficiently extends from the correct as well as the incorrect nucleotides inserted by REV1, contributing in this way to damage-induced mutagenesis. During conditions of nucleotide shortage, such as upon HU treatment in our experiments, there is no DNA lesion across which an inserter polymerase activity of REV1 would be required, which could

well explain why the REV1 catalytic activity is dispensable. Thus, Polζ may operate in a more error-free manner in fork protection, by not involving erroneous insertion of deoxycytidine (dC) by REV1.

On the other hand, the ability of Polζ to operate efficiently under nucleotide shortage in comparison to replicative polymerases[64,65] may be beneficial for cells, allowing timely completion of DNA replication under suboptimal conditions. This role of MAD2L2/REV3L could be of particular importance in cancer biology, given that nucleotide exhaustion is a common feature shared by many cellular models of oncogene overexpression and that nucleotide synthesis inhibitors are widely used in cancer therapy[66]. Moreover, biallelic mutations in *MAD2L2* have been found in patients with Fanconi Anemia, which are characterized by increased genome instability and predisposition to congenital abnormalities, bone marrow failure, and cancer. In line with the emerging view of the role of Fanconi Anemia proteins in fork protection, it is possible that compromised fork stabilization pathways, in part, contribute to tumor initiation and progression in patients harboring pathogenic mutations in *MAD2L2*.

A striking phenotype due to MAD2L2/REV3L loss is the accumulation of ssDNA in response to HU (Fig. 6b). While we have not addressed the source of the ssDNA in our context, it is plausible that the uncontrolled resection of stalled forks and defective fork restart that we observed upon MAD2L2/REV3L loss result in the exposure of ssDNA regions. We envision that MAD2L2/REV3L prevents ssDNA formation by antagonizing MRE11 activity at replication forks, and/or by promoting resynthesis of resected DNA, the latter being supported by the need for REV3L polymerase activity in fork protection (Fig. 7b). Intriguingly, this activity of MAD2L2 very much resembles the role of MAD2L2 in DSB repair, where it counteracts resection as part of shieldin, a role that appears at least in part mediated via CST-mediated recruitment of DNA polymerase α that performs fill-in synthesis[20,67–69]. However, our data does not support a role for the other shieldin factors (SHLD1, SHLD2 and SHLD3) in stalled fork protection, at least under conditions of HU-induced nucleotide depletion. This is remarkable, especially since the upstream recruiters of shieldin at DSBs, 53BP1 and RIF1, as well as the proposed downstream effector of shieldin, CST, have been implicated in counteracting nuclease-mediated resection during replication stress[10,28–32,35]. MAD2L2's ability to involve the DNA polymerase REV3L may have the advantage of allowing extensive DNA synthesis, especially when acting in complex with the accessory subunits of Polζ, POLD2 and POLD3, known to increase the efficiency and processivity of DNA synthesis by Polζ[48]. Interestingly, POLD2 and POLD3 are required for Break Induced Replication (BIR), a homologous-recombination pathway restarting stalled forks[1]. How MAD2L2 and REV3L modulate restart of HU-stalled forks is not yet clear, but given their ability to associate with POLD2/3, one plausible possibility would be that this involves BIR-mediated fork restart.

Interestingly, REV3L was recently implicated in protecting interstrand crosslink-stalled forks from EXO1 mediated degradation, by forming the protexin complex with SCAI[70]. As this role appeared not to involve MAD2L2, while we report here a MAD2L2-dependent role for REV3L in protecting HU-stalled forks from MRE11-dependent degradation, this would imply that REV3L acts in different complexes, with either SCAI or MAD2L2, to protect stalled forks against different nucleases.

Overall, our data reveal that MAD2L2/REV3L supports the progression of replication of undamaged DNA, both under unchallenged and challenged conditions, and is a critical component of the fork protection machinery. We propose a model (Fig. 7d) whereby REV1 is not essential for replication fork progression under unperturbed conditions but becomes critical under replication stress, where it supports MAD2L2/REV3L-mediated fork protection with accompanying fork restart. This function of MAD2L2/REV3L is important because it prevents genomic instability arising from DNA replication stress,

known to drive cancer development and progression. Notably, REV1-Polζ was recently identified to have a protective and mutagenic role in BRCA1/2 mutant cancer cells by filling in PRIMPOL-dependent ssDNA gaps that arise in these cells upon replication fork stalling at SMUG1-mediated DNA lesions[54]. Abolishing this gap filling role with an inhibitor that disrupts the interaction between REV1 and MAD2L2/REV3L is toxic to these HR-deficient cancer cells, including to PARP inhibitor-resistant BRCA1 mutant cells. Our identification of a role for MAD2L2/REV3L in fork protection and fork restart in absence of DNA lesions, such as upon nucleotide reduction, suggests a wider potential for MAD2L2/REV3L inhibition, or for exploiting MAD2L2/REV3L expression status, in cancer treatment, that would be worth to explore.

## Methods

### Cell culture
293 T, HeLa, BJ-hTERT and RPE1-hTERT cell lines were obtained from ATCC. RPE1-hTERT *TP53*-/- cells were a kind gift of D. Durocher[71]. All cells were maintained in Dulbecco's modified Eagle's medium (DMEM) supplemented with 10% fetal calf serum (FCS, Sigma), 100 U ml$^{-1}$ penicillin, 100 μg ml$^{-1}$ streptomycin, and 2 mM L-Glutamine (Gibco, Life Technologies), at 37 °C in a humidified atmosphere with 5% $CO_2$. All cell lines were routinely tested for mycoplasma contamination and scored negatively.

### Gene editing
For CRISPR/Cas9-mediated gene knockout, sgRNA sequences were cloned into a pLentiCRISPRv2 plasmid according to standard protocols[72]. The CRISPR-Bac system[73] was used to generate conditional knockout HeLa cells for MAD2L2, REV3L and REV1. Briefly, a piggyBac cargo vector containing a doxycycline-inducible Cas9 and hygromycin resistance gene was co-transfected with a plasmid expressing the X1-piggyBac transposase. A second cargo vector, expressing sgRNAs targeting MAD2L2, REV3L, REV1, or intergenic regions as controls, the reverse-tetracycline TransActivator (rtTA), and a gene conferring resistance to G418, was also cotransfected. A total of 10 μg of plasmid DNA at a 1:1:2 ratio of PB_rtTA_sgRNA to PB_tre_Cas9 to X1_transposase was used. Transfections were done using Lipofectamine 2000 (Thermo Fisher Scientific) according to manufacturer's instructions. Cells were subsequently selected on hygromycin (500 μg ml$^{-1}$, Gibco) and G418 (700 μg ml$^{-1}$, Gibco) for 7 to 12 days. Upon treatment with 2 μg ml$^{-1}$ doxycycline, polyclonal cell populations stably express Cas9, rtTA, and the sgRNA. All functional assays were performed after 6 days of doxycycline treatment, with the exception of survival assays and complementation experiments that involved a total of 12 days or 5 days of doxycycline treatment, respectively (see below). For lentiviral production, 293 T cells were transfected with 10 μg plasmid DNA[18]. For shieldin depletion, HeLa cells were transduced with pLentiCRISPRv2 sgRNA lentiviruses targeting *SHLD1*, *SHLD2* or *SHLD3*. Loss of protein was verified by immunoblotting when antibodies were available. Shieldin gene disruption was confirmed by PCR amplification and TIDE analysis[47] and/or quantitative real-time PCR. For gene editing information, see Supplementary Data 1.

PLentiCRISPRv2 was a gift from F. Zhang (Addgene plasmid #52961)[72]. PB_rtTA_BsmBI and PB_tre_Cas9 were gifts from M. Calabrese (Addgene plasmids #126028 and #126029)[73]. The X1-active Piggy-Bac transposase was kindly provided by L. Bombardelli (The Netherlands Cancer Institute). We are grateful to C. Canman (University of Michigan Medical School) for providing the pLLEV-EGFP-hREV1. We are grateful to L. Prakash (University of Texas Medical Branch) for providing both pCMV7-3xFLAG-zeo-hREV3L and pCMV7-3xFLAG-zeo-hREV3L D2781A/D2783A.

### Site-directed mutagenesis
To obtain the active site mutation of REV1 (residues D570A/E571A in human REV1), a site-directed mutagenesis of the pLLEV-EGFP-hREV1

vector was performed using the QuikChange II XL directed mutagenesis kit (200521, Agilent) according to manufacturer's instructions. Primers containing the desired mutations were designed using the web-based QuikChange Primer Design Program (www.agilent.com/genomics/qcpd, see Supplementary Data 2). PCR's cycling parameters were as follows: 1 cycle of 95 °C for 1 min, 18 cycles of 95 °C for 50 s, 62 °C for 50 s, and 68 °C for 12 min, and 1 cycle of 68 °C for 7 min. The PCR product was then treated with 20 U of DpnI enzyme for 2 h at 37 °C to digest the parental DNA and the mutated construct was transformed into Stbl3 *E. coli* strain (C737303, Thermo Fisher Scientific) following manufacturer's instructions. Transformed cultures were incubated overnight at 30 °C. Mutagenesis was confirmed by Sanger sequencing.

### Genomic DNA isolation and TIDE analysis

Genomic DNA was isolated using the DNeasy Blood & Tissue kit (Qiagen). PCR was performed with 200 ng DNA using Phusion High-Fidelity DNA Polymerase (Thermo Fisher Scientific) in the presence of 1% DMSO. Samples were submitted for Sanger sequencing and the editing efficiency in the different cell lines was determined using the TIDE algorithm (version 3.3.0)[47]. See Supplementary Data 2 for further details.

### RNA interference and complementation

For lentiviral production, 293 T cells were transfected with 10 μg plasmid DNA[18]. HeLa, RPE1-hTERT and BJ-hTERT cells were transduced with pLKO-puro shRNA lentiviruses from Mission library clones (Sigma) (see Supplementary Data 1 for targets and sequences). For MAD2L2 complementation experiments, HeLa cells expressing an shRNA against the MAD2L2 3' UTR were transiently transfected with pMSCV-blast retroviral vectors expressing wild-type MAD2L2 with a single FLAG-tag, described before[18], or an empty vector control. Transfections were done using Lipofectamine 2000 (Thermo Fisher Scientific) according to manufacturer's instructions. For REV1/REV3L complementation experiments, cells were first treated with 2 μg ml⁻¹ doxycycline for 5 days to induce Cas9, rtTA, and sgREV1/REV3L expression. REV1-/REV3L-depleted cells were then cultured for additional 5 days in doxycycline-free medium, supplemented with tetracycline-free FBS (A4736201, Thermo Fisher Scientific), to stop Cas9 expression. REV1-depleted cells reconstituted with REV1 were obtained by pLLEV-EGFP-hREV1 lentiviral infection. REV3L-depleted cells reconstituted with REV3L were transfected with pMSCV-zeo plasmids for transient expression of wild-type REV3L or catalytic inactive mutant (D2781A/D2783A) REV3L with a triple FLAG-tag, or with an empty pMSCV-zeo vector as a negative control. In brief, 300,000 cells were transfected with 8 μg of plasmid DNA (PEI:DNA ratio of 3:1) using linear polyethylenimine hydrochloride (PEI, 24765, Polysciences) and Opti-MEM (31985070, ThermoFisher Scientific) in antibiotic free medium. After 10 h, culture medium was refreshed to remove PEI. Cells were subsequently selected with zeocin (100 μg ml⁻¹, R25001, Thermo Fischer Scientific) for 4 days and harvested for the indicated functional assays.

Reverse siRNA transfections were done using siRNAs against MRE11 (50 nM, Silencer Select Validated, s8959, Thermo Fisher Scientific), DNA2 (40 nM, ON-TARGETplus SMARTpool, M-026431-00, Dharmacon), RAD51 (50 nM, J-003530-11-0002, Dharmacon), FBH1 (50 nM, ON-TARGETplus SMARTpool, L-017404-00-0005, Dharmacon), SMARCAL1 (50 nM, D-013058-04-0002, Dharmacon) or siRNA control against luciferase (CGUACGCGGAAUACUUCGAUU, Eurofins Genomics) with Lipofectamine RNAiMAX (Thermo Fisher Scientific) for 48 h. siRNA efficiency was analyzed by RNA extraction, reverse transcription and qPCR.

### Survival assays

To evaluate short-term survival, CRISPR-Bac cell lines were first treated with 2 μg ml⁻¹ of doxycycline for a total of 12 days to induce Cas9-mediated gene editing. On day 8, cells were treated with 1 μM CIS or 0.5 μM MMC (S8146, Selleckchem) for 4 days. Cells were then fixed with 4% formaldehyde for 20 min and stained with 0.1% crystal violet for 40 min. For quantification, crystal violet was extracted with 10% acetic acid and absorbance at 590 nm was measured in a TECAN microplate reader (Infinite M200pro, TECAN).

### Cell proliferation assays

Cell proliferation was monitored with the live cell imaging instrument IncuCyte ZOOM (Essen Bioscience). Cells were plated in a 96-well Micro Greiner clear plate (Sigma) and imaged every 4 h with default software settings and a 10x objective. The IncuCyte 2018A software was used to quantify confluence from four non-overlapping bright field images.

### Quantitative real-time PCR

To determine gene expression levels, total RNA was isolated using TRIzol (Ambion), cDNA was synthesized using the Tetro cDNA Synthesis kit (BIO-65043, Bioline) and PCR was performed using the Sensi-FAST SYBR No-ROX kit (BIO-98020, Bioline) in a QuantStudio 6 Flex real-time PCR system (Thermo Fisher Scientific). GAPDH was used as control for transcript expression. Transcripts were amplified using the primers indicated in Supplementary Data 2. The results were calculated according to the 2-ΔΔCt methodology and are shown as relative expressions to the correspondent control.

### Immunoblotting

Cells were lysed in RIPA buffer (20 mM Tris-HCl pH 7.5, 150 mM NaCl, 1 mM Na₂EDTA, 1 mM EGTA, 1 mM beta-glycerophosphate, 2 M urea, proteinase inhibitors pepstatin, leupeptin hemisulfate, aprotinin) or in 2 x SDS sample buffer, followed by a boiling step of 5 min at 95 °C. Samples were briefly sonicated for 15 s at 20% frequency. Alternatively, protein was precipitated from the organic phase of TRIzol extractions following manufacturer's instructions. Total protein concentration was measured using the Pierce BCA protein assay kit (Thermo Fisher Scientific). Equal amounts of protein were loaded onto NuPAGE Bis-Tris gels (Thermo Fisher Scientific) and electrotransferred to nitrocellulose membranes. The following antibodies were used: MAD2L2 (135977, Santa Cruz, 1:1000), REV3L (GTX100153, GeneTex, 1:500), REV1 (OTI1E12, MA5-27493, Thermo Fisher Scientific, 1:500), HSP90 (sc-7947, Santa Cruz, 1:5000), γ-Tubulin (T6557, Sigma, 1:1000), γ-H2AX (5636-I, Millipore, 1:1000), GFP (sc-8334, Santa Cruz, 1:1000), FLAG (F1804, Sigma, 1:1000), HDAC1 (PA1-860, Thermo Fisher Scientific, 1:1000), GAPDH (PA1-987, Thermo Fisher Scientific, 1:10,000). Membranes were incubated with horseradish peroxidase (HRP)-conjugated IgG secondary antibodies (1:7500 goat anti-rabbit IgG HRP G21234 or goat anti-mouse IgG HRP G21040, Thermo Fisher Scientific) or with IRDye-conjugated IgG secondary antibodies (1:10,000 IRDye 800CW donkey anti-rabbit IgG 926-32213 or IRDye 680RD donkey anti-mouse IgG 926-68072, Li-cor). Detection of chemiluminescence and fluorescence signals were carried out using enhanced chemiluminescence substrate (SuperSignal West Pico PLUS, Thermo Fisher Scientific) on a Syngene G:BOX or on a Odyssey CLx infrared scanning device (LI-COR), respectively. Blots were analyzed using either the GeneSys software (version 1.8.5.0) or the Image Studio Lite LI-COR software (version 5.2.5).

### DNA fiber assays

To study DNA replication dynamics, the DNA fiber technique was used. Exponentially growing cells were first pulse labeled with 37 μM CldU (C6891, Sigma), washed twice with phosphate-buffered saline (PBS, pH 7.4) and then labeled with 370 μM IdU (I7125, Sigma), under the conditions specified in the figure legends. Collected cells were resuspended in ice-cold PBS and DNA fiber assays were carried out as previously described[40,41]. Briefly, cells were lysed and spread on glass

slides using a spreading buffer (200 mM Tris-HCl pH 7.4, 50 mM EDTA, 0.5% SDS). DNA fibers were fixed in methanol:acetic acid (3:1), denatured in 2.5 M HCl for 30 min, and blocked in 3% Bovine serum albumin (BSA)/0.1% Tween-20 for 1 h at room temperature (RT). CldU and IdU detection was done using rat anti-BrdU antibody (Ab6326, Abcam, 1:400) and mouse anti-BrdU antibody (B44, 347580, Becton Dickinson, 1:400). Slides were fixed again with 4% formaldehyde in PBS for 10 min, followed by incubation with Alexa 568 goat antirat IgG and Alexa 488 goat anti-mouse IgG antibodies (A11077 and A11029, Thermo Fisher Scientific, 1:150) for 2 h at RT. Slides were mounted with Prolong Gold Antifade Mountant (P36930, Thermo Fisher Scientific). Fibers were imaged with a 60X objective on a Zeiss AxioObserver Z1 microscope with ZEN 2.6 software and quantified using Fiji 1.53a software. A track length of 1 μm corresponds to 2.59 kb, based on the cofactor of the 63X objective. At least 100 fibers were counted per condition.

### Immunofluorescence

For immunofluorescence to detect ssDNA-positive cells[53], cells were grown on coverslips in medium supplemented with 50 μM CldU for 16 h, followed by treatment with 4 mM HU for 2 h. Alternatively, cells were labeled with CldU and either left untreated or harvested 3 h after IR with 25 Gy. Cells were washed three times with PBS and pre-extracted with 0.5% Triton X-100/PBS on ice for 3 min. Cells were then fixed with 4% formaldehyde for 20 min at RT. After fixation, cells were washed three times with PBS, blocked with 3% BSA in PBS for 30 min at RT, and incubated with a primary antibody against CldU (ab6328, Abcam, 1:150) overnight at 4 °C. Cells were washed again three times with PBS and incubated with an IgG secondary antibody (Alexa 568 goat anti-rat IgG, A11077, Thermo Fisher Scientific, 1:100) at RT for 2 h. After washing, coverslips were mounted onto glass slides using Pro-Long Gold Antifade Mountant with DAPI (P36931, Thermo Fisher Scientific). Slides were visualized using a Zeiss AxioObserver Z1 microscope with ZEN 2.6 software at 40X magnification. At least 200 cells were counted per condition.

To monitor DNA synthesis, cells were treated with 5 μM MMC for 24 h and then labeled with 10 μM EdU for 2 h before harvesting. Alternatively, cells were either left untreated or treated with mirin (50 μM) for 2 h, followed by EdU labeling as indicated above. Cells were then washed in PBS, fixed with 3.7% paraformaldehyde in PBS for 30 min and permeabilizated with 0.5% Triton X-100 for 30 min at RT. EdU was visualized with a click-it reaction (Click-iT EdU Alexa Fluor imaging kit, Thermo Fisher Scientific) according to the manufacturer's instructions. Coverslips were mounted and imaged as indicated above. EdU integrated intensity was measured for 200 cells per condition using Fiji 1.53a software.

### SIRF assay

To examine the association of proteins to nascent DNA, the SIRF assay was performed as previously described[39], with a slight modification. Briefly, cells were grown on chamber slides (Millicell EZ slides, C86024, Millipore) and labeled with 10 μM EdU for 10 min prior to treatment with 4 mM HU for 2 h. Cells were then washed twice with PBS and permeabilized with 0.5% Triton X-100 in PBS for 10 min at 4 °C, washed twice with PBS and fixed with 3% formaldehyde, 2% sucrose in PBS for 10 min at RT. After fixation, cells were washed twice with PBS and incubated with blocking solution (3% BSA in PBS) for 30 min at RT. Slides were washed twice with PBS, followed by a Click-iT reaction (C10337, Thermo Fisher Scientific) with Biotin-azide (20 μM, Biotin-dPEG®7-azide, QBD10825, Sigma) for 30 min at RT. After two washes with PBS, primary antibodies were diluted in blocking buffer (0.02% Triton-X/5% normal goat serum/5% FCS in PBS), dispensed onto slides, and incubated overnight at 4 °C in humidified chamber. The following antibodies were used: Biotin (Jackson Immunoresearch, 200-002-211, 1:1,000), MRE11 (Novus, NB100-142, 1:100), MAD2L2 (Abcam, ab180579, 1:100), REV3L (GeneTex, GTX100153, 1:500). Slides were then washed twice with PBS,

followed by the proximity ligation assay using the Duolink PLA red reagents (DUO92002, DUO92004, DUO92008, Sigma) according to the manufacturer's instructions. Lastly, slides were mounted with DAPI-containing medium (DUO82040, Sigma) and visualized using a Zeiss LSM 980 confocal with Airyscan2 with ZEN 2.6 software at 63X magnification. At least 70 cells were analyzed per condition using ImageJ 1.53q. SIRF assays either without biotin antibody (target protein antibody only) or without EdU incubation were performed as technical negative controls to confirm the specificity of the SIRF signals.

### Neutral comet assay

To assess DNA DSBs, neutral comet assays were performed as previously described[74]. Briefly, 8000 cells were diluted in 400 μl of cold PBS and embedded in 1.2 ml 1% low-gelling agarose (Type VII-A, Sigma). 100 μl of cell suspension were then added onto Trevigen comet assay slides (4250-050-03, R&D systems) and cell lysis was performed using neutral lysis buffer (2% sarkosyl, 0.5 M Na$_2$EDTA, 0.5 mg ml$^{-1}$ proteinase K) overnight at 37 °C. The next day, slides were washed three times for 30 min with electrophoresis buffer (90 mM Tris pH 8.5, 90 mM boric acid, 2 mM Na$_2$EDTA). Electrophoresis was performed for 32 min at 20 V using the electrophoresis buffer. DNA was then stained using 2.5 μg ml$^{-1}$ of Propidium Iodide (PI) diluted in water. Slides were visualized using a Zeiss AxioObserver Z1 microscope with ZEN 2.6 software at 20X magnification. At least 50 cells were analyzed per condition using CASP software (http://www.casp.of.pl, version 1.2.3 beta 2). Neutral comet assays were performed in cells treated with 50 μM etoposide for 1 h as a technical positive control to confirm DSB detection.

### Chromosome spreads

Cells were treated with 4 mM HU for 4 h, medium was washed out and cells were allowed to grow in complete growth medium for 24 h. Before harvesting, cells were exposed to 0.2 μg ml$^{-1}$ of Colcemid (KaryoMax Colcemid Solution, Gibco BRL) for 2 h at 37 °C. Metaphases were prepared and stained with DAPI by conventional methods. Digital images of metaphases were captured using the Metafer4/MSearch automated metaphase finder system (MetaSystems) equipped with an AxioImager Z2 microscope (Carl Zeiss). Chromosomal aberrations were quantified from 50 metaphases using Fiji 2.0 software.

### Flow cytometry

For cell cycle analysis, cells were detached with trypsin/EDTA solution, washed with PBS and resuspended in ice-cold storage buffer (1% FCS/PBS) before fixation with ice-cold 70% ethanol. Cell pellets were then resuspended in 40 μg ml$^{-1}$ PI and 50 μg ml$^{-1}$ RNase A for 30 min at 37 °C. Samples were analyzed on a LSR II Fortessa Flow Cytometer (BD Bioscience). At least 10,000 cells/events were acquired for each condition, gating on live and single cells (FSC/SSC). Data was analyzed using FlowJo 10.7.1 software.

### Statistical analysis and reproducibility

Statistical analyses were performed using GraphPad Prism (version 9) and Microsoft Excel (version 16.16.27). Details on data representation, statistical tests and number of replicate experiments are indicated in the respective figure legends. Exact p-values are indicated in the figures. Additional replicates of fiber experiments and combined fiber plots are provided in the Supplementary Information file.

### Reporting summary

Further information on research design is available in the Nature Research Reporting Summary linked to this article.

## Data availability

All data generated or analyzed during this study are included in this published article and its Supplementary information files, and are

available from the corresponding author upon reasonable request. Source data are provided with this paper.

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

## Acknowledgements

We thank D. Durocher for providing the RPE1-hTERT *TP53*-/- cell line, C. Canman for the REV1 expression constructs, L. Prakash for the REV3L expression constructs, B. van den Broek from the NKI BioImaging facility for customized macros for image analysis, R. Shah for assistance with the neutral comet assay, and the members of the J. Jacobs, R. Medema and B. Rowland labs for the helpful discussions. This work was supported by project grant 11905/2018-2 from the Dutch Cancer Society (KWF) to J.J.L.J.

## Author contributions

I.P. established and validated *MAD2L2, BRCA1, SHLD2, 53BP1* and *RIF1* knockdown cell lines and *SHLD1, SHLD2* and *SHLD3* knockout cell lines, metaphase spreads, proliferation assays, DNA damage experiments, MAD2L2 complementation and RNA interference assays. Z.T. generated and validated the inducible *MAD2L2, REV3L, REV1* knockout cell lines, and performed and analyzed colony survival assays and ssDNA experiments. I.P. and Z.T. performed DNA fiber experiments and analyzed the data. M.F. performed site-directed mutagenesis, SIRF assays, generated REV1- and REV3L-depleted cells complemented with wild-type and mutant REV1/REV3L, and assisted in knockdown verification in epistasis experiments. I.P and M.F. performed and analyzed flow cytometry. Z.T. and M.F. performed and analyzed EdU incorporation assays. A.C. performed preliminary assessment of the role of MAD2L2 in replication by DNA combing. S.H.P. helped establishing the DNA fiber technique and proliferation assays and independently verified the first results of the fork degradation fiber assays in MAD2L2-depleted cells. I.P., Z.T., and J.J.L.J. wrote the manuscript. J.J.L.J. conceived the original idea, supervised the research and acquired funding.

## Competing interests

The authors declare no competing interests.
