## [Peer Review File · Nature Communications]

MAD2L2 promotes replication fork protection and recovery in a shieldin-independent and REV3L-dependent mannerEditorial Note: Parts of this Peer Review File have been redacted as indicated to maintain the confidentiality of unpublished data.

REVIEWER COMMENTS

Reviewer #1 (Remarks to the Author):

In this study, Paniagua and colleagues identify a previously unknown role for MAD2L2 in protection of replication forks, and the recovery thereof, that are stalled when cells experience (damage-independent) replication stress. In a comprehensive set of genetic experiments that encompass a wide variety of relevant gene knockouts (or knockdowns) the authors demonstrate that MRE11-dependent degradation of stalled forks is suppressed by the presence(/action) of MAD2L2, REV3 and, in part, by REV1. In this role, MAD2L2 acts independently of the binding partners with which it forms shielding, a protein complex protecting DNA breaks from resection.

As such, this work adds significantly to both the research field that focusses on the integrity of our genetic information in situations where replication is perturbed, as well as to the field studying double-strand break repair.

I find the data (*grosso modo*) convincing and supporting the conclusions. The paper is very well written and a clear demonstration of careful, thorough and meticulous experimentation. It also nicely builds on pioneering previous work of the Jacobs lab on MAD2L2. I thus support publications of this work and only have a few comments that I think may strengthen the manuscript.

*) Perhaps remarkably, I find the first experiments that is described (Fig 1a,b) the least convincing of the study. Line 100 states that MAD2L2 depletion "sensitized" ...cells to e.g. HU and aphidicolin.

However, I'm unconvinced that in the presented experiments the knocking down of MAD2L2 impact proliferation of exposed cells more than that of non-exposed cells. I would suggest rephrasing.

*) Line 124 mentions a "significant reduction". I agree the reduction is statistically significant, but consider the degree of the effect more "mild"(e.g. in clone #23) than "significant". Consider including "statistically".

*) Line 128: I suggest to remove "much" as it is rather subjective and not needed.

*) Line 134 states that "cell cycle progression remained unaffected", but is it? To me Supplemental Figure 1c is not representative for the data presented in Supplemental Figure 1d: 2 out of 3 experiments point to increased numbers of cells in G2/M. It seems to me that with such variation between 3 replicates one cannot conclude much about the cell cycle distribution. I suggest to either provide more data (e.g. other clone or more replicates) or be less definite in the language/text.

*) Later in the manuscript, data supporting an unperturbed cell cycle was presented (yet for REV3L knockdown, which, by guilt of association provides support), however, in that figure (suppl. Fig. 4) the quantification for the b panel is missing (which was helpful in figure 1).

*) To be honest, I felt Figure 4 to be quite sizable for making the point that Shieldin members are not involved in the described biology. I would consider put some of it in the supplements.

*) The only (general) concern I have with work that uses DNA fiber assays is in what way the outcomes and particularly the way these are interpreted are careful and rightful descriptions of the presumed biology - whether some effects are not confounding others. As an example, the experimental set-up described in Figure 2e has 2 periods of consecutive labelling followed by 4 hours HU exposure. A reduced IdU over CIdu ratio of know-down over control is then taken as an indication that the gene in question suppresses fork degradation. However, but perhaps I'm mistaken, Figure 2b tells me that in 4mM HU the control cells have a fork speed half of that of non-treated cells whereas the knock-outs/downs have a more dramatic reduction. To me this leads to the idea that forks still progress in the 4h timeframe in the experiment in 2d. Why is this not affecting the interpretation of the data: How can an effect on progression be separated from an interpretation of degradation? I would assume that if the forks are still able to progress in a given context, this context would be perceived as being more resilient to degradation.

- *) Line 330, Please also refer to recent work by Schimmel et al., (Nat. commun 2021).
- *) I felt lines 336-339 in the discussion section somewhat out of place, and suggest removing it altogether. I would think that if the authors wish to discuss whether CST is involved (or not, and speculate why not) they could easily experimentally address this in their study, which they chose not to. The BIR suggestion is valid without setting it against a yes or no involvement of CST.
- *) Line 358, I would suggest to remove the hyperbolic "profoundly".

All in all, very nice work!

Reviewer #2 (Remarks to the Author):

In this study, Paniagua I., et al. show that MAD2L2 and REV3 prevent nascent fork degradation by MRE11 at HU stalled replication forks and facilitate replication restart after stress. Interestingly, while this activity resembles the suppression of excessive DNA resection at DSBs by MAD2L2 within shieldin complex, this function at replication forks is not dependent on its interaction with the shieldin subunits. Instead, both TLS components, REV3L and REV1, are required for fork protection. Yet, while MAD2L2/REV3 also prevent unperturbed replication stress, REV1 is dispensable for the latter.

Overall, the study shows another nice example for the differentiation between DSB repair and replication protection. However, some of the claims are thinly supported by the data, and the overall advance would benefit from a better understanding of how these factors protect forks.

- 1) It is intriguing that REV3 and REV1 are both required for fork protection, but not Shieldin, which would be expected. So as is, it is entirely unclear how these polymerases would do so. Does it require the catalytic activities? As is, it is an interesting observation but requires more understanding to put it in the context of the scientific advance.
- 2) The current model at reversed forks is unsupported; can the authors suppress degradation with SARCAL1 KD or other reversers?
- 3) The analysis for replication restart is improper. While historically analyzed in the manner described, it since has been recognized that restart should be analyzed by keeping one of the two nucleotides with the drug, which now is the proper standard for fiber analysis. There is ample replication restart during HU, so the analysis of the dormant origin firing could be skewed. While it is not expected that this would change the overall conclusion of a restart defect in these cells here, it may make a difference for some differential analysis and is therefore important to promote proper analysis for the field.
- 4) For fibers analysis- "A representative experiment of three independent replicates is shown". So as is, only one experiment is shown, making the quantification and statistics improper. The authors should combine the data of the three experiments for representation and importantly statistical analysis.
- 5) The authors control their results in non-transformed RPE1-hTERT TP53^{-/-} cells. Why p53^{-/-}? Is the FP effect of MAD2L2 dependent on p53 and can they reproduce it in wild-type p53 cells?
- 6) Evidence that REV3/MAD2L2 but not REV1 is required for endogenous damage replication is sparse and insufficient. The fiber data in unperturbed cells does not appear to differ between shREV1 and shREV3 or shMAD2L2, but rather it differs in the control (6e: ctr is less than 1kb/min, shREV1 are slightly less than ctrl, very similar to REV3 or MAD in Fig2b, so the difference here may be the ctrl, not the KD). Again the differential functions for Rev1 and Rev3L/MAD2L2 during unperturbed replication are unclear. The authors suggest that the difference in unchallenged cells could be due to MAD2L2/REV3L but not REV1 being required for difficult to replicate regions or during transcription collisions, but show no data to support this. The authors could test this differentiated requirement by challenging the cells with G4 stabilizing drugs or transcription stalling agents to look at fibers, and so

substantiate a potential the proposed difference between REV1 and REV3/MAD2L2. Additionally, residual expression of both proteins is not shown with double knockdowns, which often are less efficient, and so can mask additive or synergistic functions and result in apparent epistasis. Can the authors use knock downs of one in a KO cell line?

7) In the introduction "nascent DNA degradation results in genomic instability, leading to Chemosensitivity" This is not correct. While restoration of fork protection defects can lead to resistance, defects in fork protection do not or barely sensitize cells to chemotherapeutics.

Additionally, the authors may consider these points:

In Fig 1d-g why is ratio >1 in NT? Can the authors express the fiber data in absolute length compared to NT?

In particular in the context of fork protection it should be discussed that MAD2L2 is a Fanconi Anemia suppressor gene.

2G: what is the significance between shMADL2L and shMAD2L2+MADL2L?

Reviewer #3 (Remarks to the Author):

In their manuscript 'MAD2L2 promotes replication fork protection and recovery in a shieldin-independent and REV3L-dependent manner', Paniagua and colleagues describe a role for MAD2L2/REV7 in the protection and restart of stalled replication forks.

MAD2L2 functions downstream of 53BP1/Rif1 in preventing the end resection of DNA breaks. In this manuscript, loss of Mad2L2 leads to defective replication, but in challenged and unchallenged cells. Subsequently, replication fork stability was measured, and a role for Mad2L2 in the protection and restart of stalled forks was observed. Surprisingly, this effects is independent on Shieldin members, but requires Rev3L/Rev1.

Overall, the phenotype of Mad2L2 is convincing, and adds another player in the increasing pallet of factors that protect stalled replication forks. With its upstream regulators Rif1/53BP1 already involved in fork protection, this role for Mad2L is not unexpected. What is surprising is that Mad2L2 protects against Mre11-mediated degradation rather than DNA2-mediated degradation, and the observation that Mad2L2 works independently of Shieldin members in this pathway. Although the genetic approaches are clear, how Mad2L2 mechanistically performs this task remains largely unclear.

Specific points:

- Is the reduced EdU incorporation in HeLa cells observed in Figure 1 due to degradation of unprotected forks? Can it be rescued by Mre11 inactivation and is similarly decreased EdU uptake seen in Rev1/3L?

- Previously RIF1 and 53BP1 were also shown to be involved in the restart and protection of stalled replication forks (Mukerjee et al, Nat Comms, 2019; Liu, Sci Adv, 2020). In those settings, fork degradation is mediated by DNA2 rather than Mre11. With Rif1/53BP1 and Mad2L2 functioning in the same pathway, it is surprising that stalled replication forks in cells lacking Mad2L2 are degraded by mre11. The authors should compare the different nucleases, and if mad2L2 indeed selectively prevents mre11-mediated degradation, the authors should address this pathway differentiation mechanistically. Also in this context, it is important that the authors verify that the experiments control for stalled forks rather than collapsed forks.

- When different players in the 53BP1/Rif1/MAD2L2 pathway protect against different nucleases (Mre11 vs DNA2), one would expect that these genes are not epistatic. As this finding is one of the novel aspects of the study, this should be addressed.

- Figure 3: the observed mitotic defects upon Mad2L2 inactivation: are these effects epistatic with loss of Rev3L/Rev1, and does Rev3L/Rev1 inactivation phenocopy Mad2L2 inactivation?

- The finding that Mad2L2 functions independently of shieldin, but dependently on REV3L/REV3 is interesting, but mechanistically unclear.

- Does Mad2L2 localize to stalled replication forks? And does it mediate the recruitment of REV3L/Rev3 to sites of fork stalling?

- Concerning the ssDNA generation in Figure 6: it is key to measure DNA break formation in these settings, to make sure that the effects reflect stalled forks, rather than resection of DSBs at collapsed forks.

- Minor: Page 5 Line 102: do -> does

Nat. Comm. manuscript: ‘MAD2L2 promotes replication fork protection and recovery in a shieldin-independent and REV3L-dependent manner’ by Paniagua et al. (corresponding author: J.J.L. Jacobs)

We thank the referees for reviewing our manuscript and appreciate their constructive comments. This has enabled us to further improve the manuscript. We hope that, with the changes made, the manuscript is now suitable for publication in Nature Communications.

Point-by-point response to remarks from the reviewers

Reviewer #1

In this study, Paniagua and colleagues identify a previously unknown role for MAD2L2 in protection of replication forks, and the recovery thereof, that are stalled when cells experience (damage-independent) replication stress. In a comprehensive set of genetic experiments that encompass a wide variety of relevant gene knockouts (or knockdowns) the authors demonstrate that MRE11-dependent degradation of stalled forks is suppressed by the presence(/action) of MAD2L2, REV3 and, in part, by REV1. In this role, MAD2L2 acts independently of the binding partners with which it forms shielding, a protein complex protecting DNA breaks from resection.

As such, this work adds significantly to both the research field that focusses on the integrity of our genetic information in situations where replication is perturbed, as well as to the field studying double-strand break repair.

I find the data (grosso modo) convincing and supporting the conclusions. The paper is very well written and a clear demonstration of careful, thorough and meticulous experimentation. It also nicely builds on pioneering previous work of the Jacobs lab on MAD2L2. I thus support publications of this work and only have a few comments that I think may strengthen the manuscript.

We are very thankful for the positive appreciation of our work and we appreciate the helpful comments that allowed us to improve the manuscript.

*) Perhaps remarkably, I find the first experiments that is described (Fig 1a,b) the least convincing of the study. Line 100 states that MAD2L2 depletion “sensitized” ...cells to e.g. HU and aphidicolin. However, I’m unconvinced that in the presented experiments the knocking down of MAD2L2 impact proliferation of exposed cells more than that of non-exposed cells. I would suggest rephrasing.

We agree with the reviewer and we have rephrased our statement in the revised manuscript. Also, we have replaced the proliferation plots to the new **Supplementary Fig. 1a**.

*) Line 124 mentions a “significant reduction”. I agree the reduction is statistically significant, but consider the degree of the effect more “mild”(e.g. in clone #23) than “significant”. Consider including “statistically”.

Thank you, we have followed this suggestion.

*) Line 128: I suggest to remove “much” as it is rather subjective and not needed.

We have changed the text.

*) Line 134 states that “cell cycle progression remained unaffected”, but is it? To me Supplemental Figure 1c is not representative for the data presented in Supplemental Figure 1d: 2 out of 3 experiments point to increased numbers of cells in G2/M. It seems to me that with such variation between 3 replicates one cannot conclude much about the cell cycle distribution. I suggest to either provide more data (e.g. other clone or more replicates) or be less definite in the language/text.

We thank the reviewer for pointing out to us that, indeed, one replicate deviated substantially from the other two replicates, complicating conclusions, and that we erroneously selected this replicate as representative example. We suspect the induction of the MAD2L2 KO went slightly different/less efficient in this one replicate. As suggested, we provided new/more data. We have performed a completely new cell cycle analysis experiment by performing three independent replicate KO induction experiments with the same control line and MAD2L2 KO line as in the original manuscript, and also with an additional control line and MAD2L2 KO line, side by side (with this whole new set of experiments also performed by a different co-author, further supporting the reproducibility towards the final conclusion).

As shown in the new **Supplementary Fig. 1e,f**, MAD2L2-depleted cells indeed accumulate in G2/M-phase, as was also seen clearly with two out of three replicates in the original manuscript (but masked by the variation caused by 1 replicate). This is consistent with a previous report (Bhat et al. 2015, PMID: 26697843) showing G2/M accumulation in the same cell type upon siRNA inhibition of MAD2L2, which was ascribed to a role for MAD2L2 in mitotic progression. In this role MAD2L2 facilitates mitotic spindle organization and chromosome segregation via its direct interaction with RAN.

Besides the new dataset, we are now showing new representative FACs plots of both control and both MAD2L2 inducible KO clones from the new experiment (new **Supplementary Figure 1e,f**). Also, the text has been adjusted accordingly.

This cell cycle analysis was primarily meant to see if there was an alteration in S-phase cells, as we see replication issues. Both the data in the original manuscript (consistent for all replicates) and the whole new experiment indicate there is no major difference in the percentage of S-phase cells.

*) Later in the manuscript, data supporting an unperturbed cell cycle was presented (yet for REV3L knockdown, which, by guilt of association provides support), however, in that figure (suppl. Fig. 4) the quantification for the b panel is missing (which was helpful in figure 1).

We apologize for this confusion, but the quantification for the **Supplementary Fig. 4b** was originally (and still is) provided in **Fig. 5a** (but **Supplementary Fig. 4b** in the original manuscript is now **Supplementary Fig. 5b** in the revision).

*) To be honest, I felt Figure 4 to be quite sizable for making the point that Shieldin members are not involved in the described biology. I would consider put some of it in the supplements.

We agree with the reviewer and we now provide a revised **Fig. 4**. We retained representative images of ssDNA-positive cells from control and shieldin-depleted cells for one sgRNA, and removed the representative images for the other sgRNAs, which significantly reduced the size of the panel **Fig. 4a**. Additionally, following suggestions from Reviewer #3, we have now included epistasis experiments with 53BP1, RIF1 and DNA2 in the new **Fig. 4**.

*) The only (general) concern I have with work that uses DNA fiber assays is in what way the outcomes and particularly the way these are interpreted are careful and rightful descriptions of the presumed biology - whether some effects are not confounding others. As an example, the experimental set-up described in Figure 2e has 2 periods of consecutive labelling followed by 4 hours HU exposure. A reduced IdU over CIdU ratio of know-down over control is then taken as an indication that the gene in question suppresses fork degradation. However, but perhaps I'm mistaken, Figure 2b tells me that in 4mM HU the control cells have a fork speed half of that of non-treated cells whereas the knock-outs/downs have a more dramatic reduction. To me this leads to the idea that forks still progress in the 4h timeframe in the experiment in 2d. Why is this not affecting the interpretation of the data: How can an effect on progression be separated from an interpretation of degradation? I would assume that if the forks are still able to progress in a given context, this context would be perceived as being more resilient to degradation.

The negative effect that MAD2L2 depletion has on fork protection can be separated from its effect in fork progression, as there is no nucleotide analogue (IdU) present during the HU treatment in the fork degradation assay (in contrast to the fork progression assay, which has a 1 h labeling period with IdU in the presence of HU). The crucial aspect here is that the labeling scheme of the fork degradation assay involves the incorporation of both nucleotide analogs prior to HU treatment. Therefore, any differences in fork progression are taken out of the equation because the analysis of fork degradation is done on the CIdU/IdU ratios, after measuring the first and second label. In this way, the fork degradation analysis already corrects for the reduced tract lengths that could be observed in cells displaying reduced fork progression rates, as it is the case for MAD2L2-depleted cells.

*) Line 330, Please also refer to recent work by Schimmel et al., (Nat. commun 2021).

We sincerely apologize for unintentionally missing this reference despite being aware of this work. We have now referenced this paper, as well as the more recent work by Paiano et al. (Genes Dev., 2021).

*) I felt lines 336-339 in the discussion section somewhat out of place, and suggest removing it altogether. I would think that if the authors wish to discuss whether CST is involved (or not, and speculate why not) they could easily experimentally address this in their study, which they chose not to. The BIR suggestion is valid without setting it against a yes or no involvement of CST.

We agree with the reviewer that this will improve the clarity of our Discussion. We have removed lines 336-339 in the revised manuscript.

*) Line 358, I would suggest to remove the hyperbolic “profoundly”.

We followed the suggestion and removed “profoundly”.

All in all, very nice work!

We very much appreciate this positive comment.

Reviewer #2

In this study, Paniagua I., et al. show that MAD2L2 and REV3 prevent nascent fork degradation by MRE11 at HU stalled replication forks and facilitate replication restart after stress. Interestingly, while this activity resembles the suppression of excessive DNA resection at DSBs by MAD2L2 within shieldin complex, this function at replication forks is not dependent on its interaction with the shieldin subunits. Instead, both TLS components, REV3L and REV1, are required for fork protection. Yet, while MAD2L2/REV3 also prevent unperturbed replication stress, REV1 is dispensable for the latter.

Overall, the study shows another nice example for the differentiation between DSB repair and replication protection. However, some of the claims are thinly supported by the data, and the overall advance would benefit from a better understanding of how these factors protect forks.

We thank the reviewer for his/her insightful and constructive comments. We have now performed multiple experiments to address these and believe that this has made our manuscript much stronger.

1) It is intriguing that REV3 and REV1 are both required for fork protection, but not Shieldin, which would be expected. So as is, it is entirely unclear how these polymerases would do so. Does it require the catalytic activities? As is, it is an interesting observation but requires more understanding to put it in the context of the scientific advance.

Indeed, this is an interesting question to address. To do so we have performed complementation experiments in REV3L- and REV1-depleted HeLa cells as indicated below:

For REV3L, we used plasmids encoding wild-type REV3L or catalytic inactive mutant (D2781A/D2783A) REV3L in Dox-induced REV3L-KO cells. As shown in new **Fig. 7b** and **Supplementary Fig. 7b**, REV3L re-expression in the REV3L-depleted cells completely rescued the extensive fork degradation, whereas expression of the catalytic inactive mutant REV3L had no effect. Thus, REV3L DNA polymerase activity is required for fork protection.

For REV1, we expressed wild-type REV1 or catalytic inactive mutant (D570A/E571A) REV1 in Dox-induced REV1-KO cells. Interestingly, both wild-type and catalytic inactive mutant REV1 rescue the fork protection defect of REV1-depleted cells, indicating that REV1's polymerase activity is not required in fork protection. These new results, included in **Fig. 7a** and **Supplementary Fig. 7a**, suggest that non-catalytic roles of REV1 are important. REV1 is generally considered to serve as a protein recruitment or landing platform to which the MAD2L2/REV3L complex (Pol ζ) binds. REV1 may thus play a more structural role in fork protection, coordinating the assembly of MAD2L2/REV3L at stalled replication forks. Moreover, REV1 catalyzes the insertion of a C opposite to a DNA lesion (whether being correct or incorrect). Subsequently, MAD2L2/REV3L can efficiently extend from the correct as well as the incorrect nucleotide inserted by REV1. In this role, REV1 and MAD2L2/REV3L play an important role in damage-induced mutagenesis. In our HU experiments there is no DNA lesion across which an inserter polymerase activity of REV1 would be required, which could well explain why the REV1 catalytic activity is dispensable. Moreover, the lack of involvement of REV1 catalytic activity may allow MAD2L2/REV3L to operate in a more error-free manner, by not involving erroneous insertion of C's by REV1.

Besides adding the new data, we have added these considerations to the discussion in the revised manuscript.

2) The current model at reversed forks is unsupported; can the authors suppress degradation with SARCAL1 KD or other reversers?

This is indeed also an interesting point to address. To support our model at reversed forks we considered the recent work from the Cortez lab (Liu et al., 2020), which suggests that there are (at least) two remodeling pathways to generate the DNA substrates for nucleolytic degradation at stalled forks. Both mechanisms are dependent on RAD51 and differ on whether they use FBH1 or the SMARCAL1, ZRANB3 and HLTF motor proteins. In the revised manuscript, we performed fork degradation assays in control and MAD2L2-depleted cells upon siRNA-mediated downregulation of RAD51, SMARCAL1 and FBH1, with the goal of targeting the common upstream factor (RAD51) and one of the components of each remodeling pathways (FBH1, SMARCAL1). Our data shows that depletion of both RAD51 and SMARCAL1, but not FBH1, completely rescues nascent strand degradation in MAD2L2-depleted cells (new **Fig. 3a** and **Supplementary Fig. 3a-d**). Consistent with earlier published reports that RAD51 is essential for replication fork reversal (Zellweger et al. (J Cell Biol, 2015), Mijic et al. (Nat Comm, 2017), Bhat et al. (Cell Reports, 2018) and that SMARCAL1 is a DNA translocase with demonstrated activity in fork reversal (Betous et al. (Genes and Dev, 2012)), these results are a very strong indication that fork reversal is indeed required for nascent DNA degradation in MAD2L2-depleted cells. We would like to also point out that FBH1 generates the DNA substrate that is degraded when 53BP1 is inactivated (Liu et al., 2020). Given the non-epistatic relationship between 53BP1 and MAD2L2 in fork protection that we now describe in the revised manuscript (see response to reviewer #3), the lack of requirement for FBH1 in fork reversal in the absence of MAD2L2 further reinforces our model, that the role of MAD2L2 at stalled forks is different from its role at DSBs, where it requires 53BP1 and shieldin.

3) The analysis for replication restart is improper. While historically analyzed in the manner described, it since has been recognized that restart should be analyzed by keeping one of the two nucleotides with the drug, which now is the proper standard for fiber analysis. There is ample replication restart during HU, so the analysis of the dormant origin firing could be skewed. While it is not expected that this would change the overall conclusion of a restart defect in these cells here, it may make a difference for some differential analysis and is therefore important to promote proper analysis for the field.

We thank the reviewer for pointing this out, we were not aware of changes in the standards for fiber analysis, as most recent replication studies from high-profile labs (Rageul et al. (Nat Comm, 2020), Shun Yu Lo et al., (Sci Adv, 2021), Tian et al. (Mol Cell, 2021), Genois et al. (Mol Cell, 2021)) and unpublished work in conferences analyze fork restart as we have done. Following the reviewer's suggestion, we now quantified fork restart efficiency in one control and two MAD2L2-depleted cell lines, whereby CldU pulse-labeled forks under 1h HU treatment were measured (the first analogue CldU is kept with the HU drug) to determine whether they can restart after HU wash off (IdU pulse-label). Similar to what we determined with the conventional fork restart assay, MAD2L2-depleted cells show defective fork restart, increased percentage of stalled forks and firing of dormant origins. These results are presented in the new **Supplementary Fig. 2a**.

4) For fibers analysis- "A representative experiment of three independent replicates is shown". So as is, only one experiments is shown, making the quantification and statistics improper. The

authors should combine the data of the three experiments for representation and importantly statistical analysis.

In the main figures of the original manuscript, we show the variation between measurements from a single representative fiber experiment, with at least 100 fibers per condition and per replicate. We also provided the information from the other two out of three biological replicates in **Supplementary Data file 1** as figures with statistics clearly visible, to make all fiber data easily available to the reader. We agree that, in general, the most relevant information for the reader is the variation between biological replicates. However, data generated from DNA fiber stainings, in terms of DNA tract lengths, vary significantly from one experiment to the next, making even relatively strong effects difficult to evaluate for statistical significance. We would preferably show a single experiment with statistics on the variation between measurements in the main figures, with the other experiments in the supplements as we did, as it is the common practice among our peers in the field (Chaudhuri et al. (Nature, 2016), Mijic et al. (Nat Comm, 2017), Maya-Mendoza et al. (Nature, 2018), Mukherjee et al. (Nat Comm, 2019), Adeyemi et al. (Molecular Cell, 2021), Lyu et al. (EMBO, 2021), Tagliatela et al. (Molecular Cell, 2021)). In addition to the revised **Supplementary Data file 1**, we now included a **Source Data file** with all individual data points from the three biological replicates.

Independent additional motivation for our choice to show all individual data, instead of pooling the data, comes from that we noticed in the published peer review file associated with Rageul et al, 2020 (Nature Comms., PMID: 33127907), a reviewer specifically requested to change the data representation from pooled data to showing the individual experiments, as we did (with one representative experiment and the others in supplemental data).

5) The authors control their results in non-transformed RPE1-hTERT TP53^{-/-} cells. Why p53^{-/-}? Is the FP effect of MAD2L2 dependent on p53 and can they reproduce it in wild-type p53 cells?

There was no particular reason for choosing a p53-deficient RPE1-hTERT TP53^{-/-} cell line to control our results. Our goal was to validate our results in a non-transformed cell line that proliferates well and tolerates MAD2L2 depletion, given that different cell lines differ in how well they tolerate a strong MAD2L2 depletion. Following the reviewer's suggestion, we now also performed fork degradation assays in two additional non-transformed cell lines that are wild-type for p53: in p53-proficient BJ-hTERT cells as well as in p53-proficient RPE1-hTERT cells.

Further increasing the rigor of our study by showing the same phenotype in multiple cell line backgrounds, fork degradation assays in non-transformed p53-proficient BJ-hTERT cells yielded similar results as we observed in HeLa and RPE1-hTERT TP53^{-/-} cells: knocking down MAD2L2 with two independent shRNAs significantly reduces the IdU/CldU ratios, indicating that MAD2L2 is required for efficient fork protection in both noncancerous and cancer cell lines. These results are included as new **Fig. 2f** and **Supplementary Fig. 2f**.

However, unexpectedly, in contrast to RPE-hTERT TP53^{-/-} cells, loss of MAD2L2 in p53-proficient RPE1-hTERT cells did not result in nascent tract degradation (**Supplementary Fig. 2g,h**). This finding may reflect a previously proposed role for p53 in fork protection (Hampp et al., (PNAS, 2016)), or alternatively an involvement of p53 in the remodeling of stalled forks (Hampp et al., (PNAS, 2016) and Roy et al., (eLife, 2018)). As we did observe fork degradation in p53-proficient BJ-hTERT cells depleted for MAD2L2, it is thus possible that the role of p53 in replication fork stabilization is cell-type specific. Something that we feel is beyond this study to further dive into.

6) Evidence that REV3/MAD2L2 but not REV1 is required for endogenous damage replication is sparse and insufficient. The fiber data in unperturbed cells does not appear to differ between shREV1 and shREV3 or shMAD2L2, but rather it differs in the control (6e: ctr is less than 1kb/min, shREV1 are slightly less than ctrl, very similar to REV3 or MAD in Fig2b, so the difference here may be the ctrl, not the KD).

We thank the reviewer for noticing this discrepancy, which led us to revisit this point and have a closer look at our fiber data. As mentioned above, there are small technical variations in the measured DNA tract lengths per experiment. We have therefore shown all three independent replicates (one in the main figure, the other two in the **Supplementary Data file 1**). The phenotypes are very consistent among the replicates. We have perhaps not chosen the best representative one out of the three replicates for **Fig. 2b**, since the fork progression rates for the untreated control cells are a bit higher in this experiment than in the other replicates. We have now selected for **Fig. 2b** a similar panel from the **Supplementary Data file 1**, where the difference between control and MAD2L2-depleted cells is more evident.

Again the differential functions for Rev1 and Rev3L/MAD2L2 during unperturbed replication are unclear. The authors suggest that the difference in unchallenged cells could be due to MAD2L2/REV3L but not REV1 being required for difficult to replicate regions or during transcription collisions, but show no data to support this. The authors could test this differentiated requirement by challenging the cells with G4 stabilizing drugs or transcription stalling agents to look at fibers, and so substantiate a potential the proposed difference between REV1 and REV3/MAD2L2.

[redacted]

Additionally, residual expression of both proteins is not shown with double knockdowns, which often are less efficient, and so can mask additive or synergistic functions and result in apparent epistasis. Can the authors use knock downs of one in a KO cell line?

Less efficient knockdown can indeed be a risk, in particular when knocking down both factors through the same mechanism as RNAi. However, we do not use double knockdown by RNAi in our experiments. Indeed, in all our epistasis experiments we use a combination of inducible CRISPR knockout of one factor and knockdown by RNAi of the other factor, or we use a pharmacological inhibitor. We make use of a well-established inducible KO system, as full KO clones for multiple factors in our study are not viable (e.g. REV3L) or affected in growth, and we do our experiments in a carefully timed short-term manner to avoid potential selection. Protein/RNA expression data from epistasis experiments are included in **Supplementary Figures 3, 4, 6**, of the revised manuscript.

We see no indications for less efficient depletion of factors by RNAi in double vs single depletion conditions.

7) In the introduction “nascent DNA degradation results in genomic instability, leading to Chemosensitivity“ This is not correct. While restoration of fork protection defects can lead to resistance, defects in fork protection do not or barely sensitize cells to chemotherapeutics.

We thank the reviewer for the clarification. We have modified this statement in the revised manuscript, which now appears as follows: “*Consequently, in the absence of key protective*

factors such as BRCA1/2, FANCD2, FANCA and ABRO1, nascent DNA degradation results in genomic instability, an enabling hallmark of cancer”.

Additionally, the authors may consider these points:

In Fig 1d-g why is ratio >1 in NT? Can the authors express the fiber data in absolute length compared to NT?

We think the reviewer refers to the fork degradation assays in **Fig. 2d, 2g** of the original manuscript (**Fig. 2e, 2i** now in the revised manuscript). An IdU/CldU ratio >1 for the control condition in these figures and also in several other fork degradation assays in the manuscript, while in others the ratio is (close to) 1, is due to incorporation of the second label (IdU) having continued a bit longer than that of the first label (CldU), and relates to technical variability between experiments. The concentration of the second nucleotide analogue (IdU) is 10-fold higher than that of the first analogue (CldU). This is to ensure that the second analogue can displace any CldU that was not removed during the washing steps, providing an efficient block to further incorporation of the first label. Incorporation of the second analogue is only countered by washing it out at the end of the incubation time. The procedure of adding and washing the analogues from the cell culture must be performed in a precise and quick manner to ensure accuracy of the incorporation within the labeling window. If the second label (IdU) is not washed out completely before starting the HU treatment (or if there would be slightly longer incubation of the second label (IdU) than of the first label (CldU)), the IdU tract lengths are longer compared to the CldU tracts, leading to IdU/CldU ratios >1.

In this project two researchers have independently performed and analyzed fork dynamics assays from hundreds of samples. Unfortunately, the small technical variability that results from processing experiments independently is reflected in the data. Despite such technical variability, both researchers have successfully extracted novel biological patterns from multiple fiber experiments without applying correction methods. We believe that attempting to correct for these sources of variability will do more harm than good. Important of course is that in every individual experiment all conditions were processed together and in no specific order by the same investigator, so any variation in washout or incubation time, applies to all conditions within an experiment.

We prefer to show the fiber ratios as this is consistent with the currently most common way of presenting fiber data. Plotting absolute fiber lengths instead of ratios is not changing anything about the second label tract length being longer than the first label tract length due to e.g. a less good wash out and/or slightly longer incubation of the second label.

Nevertheless, for the referee we have plotted below the experiment shown in manuscript **Fig. 2e** in absolute tract lengths in μm .

Figure 2. Absolute fiber tract length in μm for the experiment shown in Fig. 2e. Shown are the lengths of the individual CldU (1^{st} label) and IdU tract (indicative of the degradation) and the total fiber tract length (IdU plus CldU). For values in kb, multiply by 2,59.

In particular in the context of fork protection it should be discussed that MAD2L2 is a Fanconi Anemia suppressor gene.

We thank the reviewer for this suggestion and we have made the necessary changes in the Discussion section to highlight the participation of MAD2L2 in the Fanconi Anemia pathway and the possibility that the novel role of MAD2L2 in fork protection hereby described could also contribute to the suppression of the Fanconi Anemia phenotype.

2G: what is the significance between shMADL2L and shMAD2L2+MADL2L?

We now included this statistical comparison.

Reviewer #3

In their manuscript ‘MAD2L2 promotes replication fork protection and recovery in a shieldin-independent and REV3L-dependent manner’, Paniagua and colleagues describe a role for MAD2L2/REV7 in the protection and restart of stalled replication forks. MAD2L2 functions downstream of 53BP1/Rif1 in preventing the end resection of DNA breaks. In this manuscript, loss of Mad2L2 leads to defective replication, but in challenged and unchallenged cells. Subsequently, replication fork stability was measured, and a role for Mad2L2 in the protection and restart of stalled forks was observed. Surprisingly, this effects is independent on Shieldin members, but requires Rev3L/Rev1. Overall, the phenotype of Mad2L2 is convincing, and adds another player in the increasing pallet of factors that protect stalled replication forks. With its upstream regulators Rif1/53BP1 already involved in fork protection, this role for Mad2L is not unexpected. What is surprising is that Mad2L2 protects against Mre11-mediated degradation rather than DNA2-mediated degradation, and the observation that Mad2L2 works independently of Shieldin members in this pathway. Although the genetic approaches are clear, how Mad2L2 mechanistically performs this task remains largely unclear.

We thank the reviewer for the helpful comments. By addressing these below, as well as by addressing the comments from the other reviewers, we have performed significant additional analysis on the function of MAD2L2/REV3L/REV1 at stalled replication forks. We hope that our effort is sufficient to convince the reviewer.

Specific points:

- Is the reduced EdU incorporation in HeLa cells observed in Figure 1 due to degradation of unprotected forks? Can it be rescued by Mre11 inactivation and is similarly decreased EdU uptake in Rev1/3L?

We performed the experiment indicated by the reviewer, addressing if rescuing fork degradation by Mirin would rescue the reduced EdU incorporation of MAD2L2-depleted cells (in absence of exogenous damage). We present the results here as **Figure 3** for the reviewer.

Figure 3. Assessment of DNA synthesis rates in control and MAD2L2-depleted cells upon MRE11 inactivation.

In the control cells Mirin treatment resulted in a considerable increase in EdU incorporation. We think this might reflect that control HeLa cells already have elevated replication stress, including a high proportion of stalled replication forks that undergo degradation. In line with this, SIRF assays added to the revised manuscript (new **Fig. 1f**) detected association of MRE11 at nascent DNA already in DMSO conditions, which was strongly enhanced upon HU, in line with published SIRF data for MRE11. Blocking MRE11 activity with Mirin in this setting may rescue stalled fork degradation, allowing the reinitiation of DNA synthesis and thus the increase EdU incorporation.

On the other hand, the reduced EdU incorporation rates in the MAD2L2-depleted cells under unchallenged conditions were only slightly improved/rescued by Mirin (and nowhere near that of Mirin treated control cells).

We hypothesized that the reduced EdU incorporation in MAD2L2-depleted cells results from fork stalling at endogenous lesions or other (structural) barriers to replication. In line with this, the SIRF experiments added to the revised manuscript (new **Fig. 1g**) indicate association of MAD2L2 with nascent DNA under unchallenged (DMSO) conditions.

Part of the endogenous causes of fork stalling might need MAD2L2 (in Polζ) for continuation of replication past these endogenous replication blocking lesions/structures. A rescue of fork degradation alone would not overcome this and restore EdU incorporation.

Moreover, we have shown in this manuscript that MAD2L2-depleted cells not only have a fork degradation defect but also have a defect in replication fork restart (with HU). Thus, even if Mirin treatment prevents the aberrant processing of stalled forks, replication forks may not be able to restart efficiently in the absence of MAD2L2, explaining the poor rescue of the EdU incorporation rates upon Mirin in these cells.

Regarding the second question, we believe the reviewer might have overlooked that data showing the effect of REV3L and REV1 depletion on EdU incorporation rates, were/are included in the manuscript in **Fig. 5b** and **Fig. 6d**, respectively. Indeed, we see similar decreased EdU incorporation rates after mitomycin C treatment between MAD2L2-, REV3L- and REV1-depleted cells, while in unchallenged conditions we see reduced EdU incorporation for MAD2L2 and REV3L-depleted cells, but not REV1-depleted cells (please see manuscript text for further comments).

(As a small side note: we noticed a small labeling mistake of the control cells in the original **Fig. 1e**, that we now corrected. The representative images and quantification are shown for the same control cells (not one with a CTRL #2 and other with a CTRL #1, as the old labeling erroneously indicated)).

- Previously RIF1 and 53BP1 were also shown to be involved in the restart and protection of stalled replication forks (Mukerjee et al, Nat Comms, 2019; Liu, Sci Adv, 2020). In those settings, fork degradation is mediated by DNA2 rather than Mre11. With Rif1/53BP1 and Mad12 functioning in the same pathway, it is surprising that stalled replication forks in cells lacking Mad2L2 are degraded by mre11. The authors should compare the different nucleases, and if mad2l2 indeed selectively prevents mre11-mediated degradation, the authors should address this pathway differentiation mechanistically.

We appreciate this suggestion from the reviewer. This is indeed an interesting point to address experimentally. RIF1, 53BP1 and MAD2L2 indeed function in the same pathway, but this is in the context of DSB repair. At stalled replication forks, RIF1 is already shown to act independently (Mukerjee et al. (Nat Comm, 2019), Alabert et al. (Nat Cell Bio, 2014)). For 53BP1, although the data is more conflicting, it has been suggested to not require RIF1 in its fork protection activity (Liu et al. (Sci Adv, 2020), Chaudhuri et al. (Nature, 2016)). We now performed additional fork degradation assays that further support this pathway differentiation. Unlike the case for MRE11, siRNA-mediated depletion of the DNA2 nuclease does not rescue fork degradation in MAD2L2-depleted cells (new **Fig. 4f**), suggesting that MRE11 plays a more dominant role in nucleolytic degradation of stalled forks in MAD2L2-depleted cells. In addition, we have performed epistasis analysis of MAD2L2 depletion with RIF1 and 53BP1 depletion, as well as investigated the involved fork remodelers, to obtain more insight in this pathway differentiation. Please see new **Fig. 3a** and **4e** and their discussion in the manuscript text.

Also in this context, it is important that the authors verify that the experiments control for stalled forks rather than collapsed forks.

To address this comment, we first performed fork degradation assays using a low dose of HU (300 μ M), which has been reported to cause replication fork stalling, but not replication fork collapse (Benedict et al., Life Sci Alliance, 2018; Roy et al., Elife, 2018). As shown in new **Supplementary Fig. 2i**, the fork degradation phenotype associated with MAD2L2 depletion is also observed with the low dose of HU. Although we cannot formally rule out the formation of a few DSBs with the low dose of HU, we conclude that MAD2L2 is indeed acting at stalled forks rather than at broken forks. Furthermore, to determine whether the dose of HU used in the fork degradation assays (4 mM HU) leads to the formation of broken forks – that is, DSBs –, we examined DSB formation by neutral comet assays. Control and MAD2L2-depleted cells

did not show major changes in DSB induction after treatment with 4 mM HU for 4 h compared to the untreated condition. These new results are included in new **Fig. 3g,h**.

- When different players in the 53BP1/Rif1/MAD2L2 pathway protect against different nucleases (Mre11 vs DNA2), one would expect that these genes are not epistatic. As this finding is one of the novel aspects of the study, this should be addressed.

This is an excellent point. To address epistasis in fork protection, we performed fork degradation assays in control, 53BP1- and RIF1-depleted cells, with or without inactivation of MAD2L2 (results are depicted in the new **Fig. 4e and Supplementary Fig. 4d-f**). We found that while MAD2L2 depletion resulted in increased fork degradation rates upon HU, both in control and 53BP1-depleted cells, MAD2L2 depletion did not further increase fork degradation rates upon HU in RIF1-depleted cells. This indicates that MAD2L2 acts in fork protection in a different pathway than 53BP1, but has an epistatic relationship with RIF1 in fork protection.

The non-epistatic relationship between 53BP1 and MAD2L2 in fork protection is in line with the lack of requirement for FBH1-mediated fork reversal in MAD2L2-depleted cells, as FBH1 is required for fork degradation when 53BP1 is inactivated (Liu et al., Sci Adv, 2020). Furthermore, MAD2L2 does not protect against excessive DNA2 activity at stalled forks, while 53BP1 does.

On the other hand, the epistatic relationship between RIF1 and MAD2L2 in fork protection is relatively unexpected, when looking from a straightforward or simple perspective. The mechanistic basis for this is unclear at this point, but the explanation might lie in the order of activity or substrate specificities of the different nucleases that attack stalled forks and in how the forks have been remodeled. Stalled forks processed by MRE11 in absence of MAD2L2 may be prohibited from further processing by other nucleases such as DNA2, or may not be substrates of RIF1-mediated protection, causing additional depletion of RIF1 to be without effect in MAD2L2-depleted cells. Of note, it is not known on which fork remodelers the nascent strand degradation in RIF1-depleted cells depends.

- **Figure 3: the observed mitotic defects upon Mad2L2 inactivation: are these effects epistatic with loss of Rev3L/Rev1, and does Rev3L/Rev1 inactivation phenocopy Mad2L2 inactivation?**

To address this comment, we have assessed chromosomal aberrations in metaphase spreads of cells depleted for REV1 or REV3, alone or in combination with MAD2L2 depletion (see new **Fig. 7c, Supplementary Fig. 7c**). Already under unchallenged (DMSO) conditions, REV3L loss showed elevated chromosomal aberrations, which was not observed for REV1. This is consistent with reported findings by Bhat et al. (NAR, 2013 PMID: 23303771) and Lange et al. (NAR, 2012 PMID: 22319213). The levels of chromosomal aberrations clearly increased when REV3L-depleted or MAD2L2-depleted cells, but not REV1-depleted cells, were challenged with HU, and included chromatid and chromosome breaks, but also radial figures that are believed to originate from toxic repair activities. This effect was exacerbated when MAD2L2 was co-depleted in REV3L-depleted cells. The chromosomal aberrations in REV3L-depleted cells have previously been linked to a role for REV3L in maintaining common fragile sites, which represent 'hot spots' of chromosomal breakage.

The absence of increased chromosomal aberrations in REV1-depleted cells, suggests that the genomic loci with increased fork degradation and ssDNA generation upon HU are mostly restored correctly in these cells. On the other hand, additional roles of REV3L and MAD2L2, such as in fragile site stability and in DNA repair, respectively, may contribute to chromosomal aberrations in REV3L- and MAD2L2-depleted cells. The additional increase in chromosomal

aberrations when MAD2L2 is depleted on top of REV3L might reflect the additional role of MAD2L2 in DSB repair.

- The finding that Mad2L2 functions independently of shieldin, but dependently on REV3L/REV3 is interesting, but mechanistically unclear.

In our efforts to address the comments of all reviewers, we have performed significant additional experiments for the revision of our manuscript that provide more mechanistic insight.

These include the localization of MAD2L2 and REV3L to sites of stalled replication (SIRF assays), revealing the involvement of the REV3L catalytic activity, but not the REV1 catalytic activity, the requirement of MAD2L2 for the recruitment of REV3L to sites of HU-stalled replication, experiments addressing the type of remodeled fork that MAD2L2 acts on to protect against degradation, epistasis analysis with 53BP1 and RIF1, and revealing that MAD2L2 does not protect against DNA2 mediated fork degradation.

In addition, previous work on the mechanisms of MAD2L2 interactions with its different partners (extensively summarized and discussed in our review, de Krijger et al., Trends Cell Biol. 2021) have revealed how REV3L and SHLD3 each interact with MAD2L2. In the context of DSBs, MAD2L2 and 53BP1/RIF1 are bridged by SHLD3. SHLD3 and REV3L form mutual exclusive interactions with the seatbelt element of MAD2L2. Given that SHLD3 depletion does not affect replication fork protection (**Fig. 4d**), we believe that SHLD3 is not involved in recruiting MAD2L2 to replication forks but that rather REV1, the recruiter of Pol ζ in TLS, localizes MAD2L2/REV3L to suppress ssDNA formation. In line with our results that the catalytic activity of REV1 is dispensable.

For further discussion of these different mechanistic aspects, we refer to the manuscript.

- Does Mad2L2 localize to stalled replication forks? And does it mediate the recruitment of REV3L/Rev3 to sites of fork stalling?

We have now performed proximity ligation assays (PLA) with EdU and MAD2L2/REV3L. The technique is also known as *in situ* protein interaction with nascent DNA replication forks (SIRF). We have included a MRE11-EdU SIRF experiment as a positive control for the SIRF assay. The association of MAD2L2 to nascent DNA was detected in unchallenged conditions and further increased upon HU treatment, confirming that indeed MAD2L2 localizes to stalled replication forks. We also detected localization of REV3L to nascent DNA upon HU treatment, that was dependent of MAD2L2. These results are included as new **Fig. 1f,g** and **Fig. 5f**.

- Concerning the ssDNA generation in Figure 6: it is key to measure DNA break formation in these settings, to make sure that the effects reflect stalled forks, rather than resection of DSBs at collapsed forks.

Since the conditions of HU treatment were quite similar between the fork degradation assay (4 mM HU, 4 h) and the ssDNA generation assay (4 mM HU, 2h), we performed neutral comet assays to measure DSB formation upon treatment with 4 mM HU for 4 h. These assays did not reveal an increase in DSBs upon HU, nor any differences between control and MAD2L2-depleted cells, and are included in the revised manuscript in new **Fig. 3g,h**.

- Minor: Page 5 Line 102: do -> does.]

In line 102, 'do' refers to 'agents', which is plural, so we think it should be 'do', not 'does'.

REVIEWERS' COMMENTS

Reviewer #1 (Remarks to the Author):

The authors did a truly excellent job in improving their already high-quality and original manuscript, not only by satisfactorily addressing the points I made in the original review but also by including an impressively great number of new experiments following the suggestions made by all reviewers: e.g. demonstrating that the enzymatic activity of REV3 but not of REV1 (Ref#2) is needed is a substantial step forward towards mechanistic insight into how REV1/3/7 counteracts potential fork degradation, so is the demonstration of an epistatic relation between REV7 and RIF1 (yet not of REV7 and 53BP1) (Ref#3), and the dependency of nascent DNA degradation in MAD2L2-depleted cells on SMARCAL1 (Ref#2).

I thus strongly support publication of this elegant and impactful study

Reviewer #2 (Remarks to the Author):

Overall the authors have addressed all comments, and added important data that strengthens the manuscript.

For the authors consideration in moving forward with fiber analysis beyond this manuscript, a good review of the issues and differential information obtained by different labeling schemes is <https://www.sciencedirect.com/science/article/pii/S0076687917301143?via=ihub> although there are certainly others e.g. by Pasero et al. The authors pointed to many publications that use fiber analysis, which is exactly the point why it is important to perpetuate proper analysis and experimental set up, as differential labeling (first label with HU, second label with HU, no label with HU) address different aspects of restart/progression/origin usage.

Another point to consider: while I agree with the authors that in all biological replicas, the pattern is consistent with the notion that MAD2L2 KD enhances an intrinsic fork protection of a given cell line, and it is fine to show the individual plots to make this point, it should not replace a pooled data plot: Mathematically, by definition increasing the data points and including both biological and technical replicates will increase the significance of any difference. If it doesn't, it that means that there are other factors than the genotypes under investigation that are skewing the individual results.

Reviewer #3 (Remarks to the Author):

The authors have extensively addressed my comments, and convincingly demonstrate a role for MAD2L2 in fork processing, which is mechanistically separate from its role at DSBs. I endorse publication.

In the rebuttal letter, the authors show an interesting experiment on MRE11 inhibition on replication in control and Mad2L2-defective cells, also the corresponding discussion is relevant. I would suggest to include that experiment in the supplemental figures of the experiment.

Nat. Comm. manuscript: ‘MAD2L2 promotes replication fork protection and recovery in a shieldin-independent and REV3L-dependent manner’ by Paniagua et al. (corresponding author: J.J.L. Jacobs).

Point-by-point response to remarks from the reviewers

Reviewer #1

The authors did a truly excellent job in improving their already high-quality and original manuscript, not only by satisfactorily addressing the points I made in the original review but also by including an impressively great number of new experiments following the suggestions made by all reviewers: e.g. demonstrating that the enzymatic activity of REV3 but not of REV1 (Ref#2) is needed is a substantial step forward towards mechanistic insight into how REV1/3/7 counteracts potential fork degradation, so is the demonstration of an epistatic relation between REV7 and RIF1 (yet not of REV7 and 53BP1) (Ref#3), and the dependency of nascent DNA degradation in MAD2L2-depleted cells on SMARCAL1 Ref#2).

I thus strongly support publication of this elegant and impactful study.\

Reviewer #2 (Remarks to the Author):

Overall the authors have addressed all comments, and added important data that strengthens the manuscript.

For the authors consideration in moving forward with fiber analysis beyond this manuscript, a good review of the issues and differential information obtained by different labeling schemes is <https://www.sciencedirect.com/science/article/pii/S0076687917301143?via=ihub>

although there are certainly others e.g. by Pasero et al. The authors pointed to many publications that use fiber analysis, which is exactly the point why it is important to perpetuate proper analysis and experimental set up, as differential labeling (first label with HU, second label with HU, no label with HU) address different aspects of restart/progression/origin usage. Another point to consider: while I agree with the authors that in all biological replicas, the pattern is consistent with the notion that MAD2L2 KD enhances an intrinsic fork protection of a given cell line, and it is fine to show the individual plots to make this point, it should not replace a pooled data plot:

Mathematically, by definition increasing the data points and including both biological and technical replicates will increase the significance of any difference. If it doesn't, it that means that there are other factors than the genotypes under investigation that are skewing the individual results.

Reviewer #3

The authors have extensively addressed my comments, and convincingly demonstrate a role for MAD2L2 in fork processing, which is mechanistically separate from its role at DSBs. I endorse publication.

In the rebuttal letter, the authors show an interesting experiment on MRE11 inhibition on replication in control and Mad2L2-defective cells, also the corresponding discussion is relevant. I would suggest to include that experiment in the supplemental figures of the experiment.

We thank the reviewers for their constructive criticism and positive evaluation of our revised manuscript. We have now added the combined plots for all fiber replicates to the Supplementary Data file 1 and included the mirin EdU experiment (Supplementary Figure 3g) and discussion.